# No Alignment Needed for Generation: Learning Linearly Separable Representations in Diffusion Models

## Abstract

Efficient training strategies for large-scale diffusion models have recently emphasized the importance of improving discriminative feature representations in these models. A central line of work in this direction is representation alignment with features obtained from powerful external encoders, which improves the representation quality as assessed through *linear probing*. Alignment-based approaches show promise but depend on large pretrained encoders, which are computationally expensive to obtain. In this work, we propose an alternative regularization for training, based on promoting the **L**inear **SEP**arability (LSEP) of intermediate layer representations. LSEP eliminates the need for an auxiliary encoder and representation alignment, while incorporating linear probing directly into the network's learning dynamics rather than treating it as a simple post-hoc evaluation tool. Our results demonstrate substantial improvements in both training efficiency and generation quality on flow-based transformer architectures such as SiTs, achieving an FID of 1.44 on $256 \times 256$ ImageNet and FID of 1.66 on $512 \times 512$ ImageNet dataset.

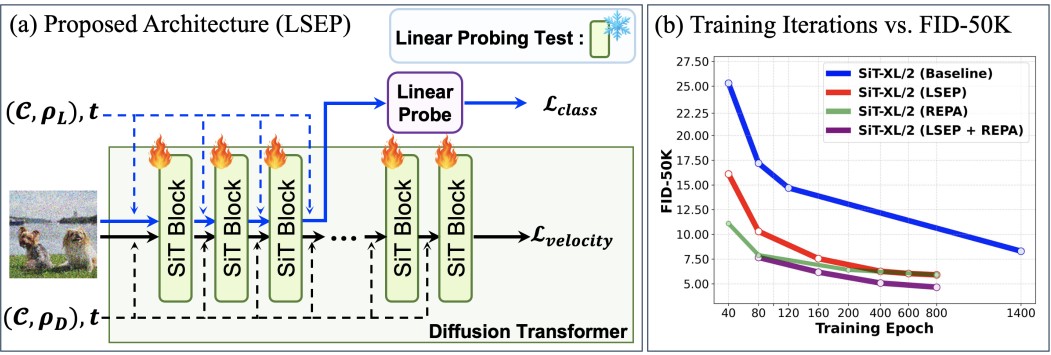

Figure 1: (a) Overview of the proposed *Linear SEParability (LSEP)* regularization. Unlike the linear probing test (Alain & Bengio, 2016), where separability of a target layer is evaluated on a frozen model, LSEP unlocks the layers and jointly optimizes a linear probe branch/classifier along with the denoising process. This actively drives the target layers toward *higher linear separability* of features, substantially enhancing the generative model's effectiveness on the denoising task *without relying on any large-scale external encoder*. (b) SiT-XL with LSEP exhibits markedly faster FID improvement (without classifier-free guidance) than baseline SiT-XL, and by 4M iterations, it *achieves an FID score equivalent* to that of the alignment-based REPA model.

## 1 Introduction

Diffusion models have demonstrated remarkable generative quality in a wide range of visual tasks, motivating additional architectural innovations to further improve their performance (Dhariwal & Nichol, 2021; Karras et al., 2022; Peebles & Xie, 2023; Ma et al., 2024). These models inherently learn semantically meaningful representations through the noise-prediction objective used in denoising (Baranchuk et al., 2022; Xiang et al., 2023; Chen et al., 2025). However, their learned

representations are not explicitly optimized for representation learning and are often less expressive than models trained specifically for this purpose (Xiang et al., 2023; Yu et al., 2025).

From a representational perspective, a growing body of research has examined how improving learned representations can benefit diffusion training (Zhu et al., 2024; Yu et al., 2025; Jiang et al., 2025; Leng et al., 2025; Yao et al., 2025). These studies have explored methods to improve the training efficiency and generative performance of transformer-based diffusion models through *representation alignment*. In particular, they have consistently shown that acquiring strong representations at specific network depths by aligning with high-quality internal (Zhu et al., 2024; Jiang et al., 2025) or external (Yu et al., 2025; Leng et al., 2025; Yao et al., 2025) representations significantly accelerates convergence and enhances the quality of generated outputs.

A recent powerful approach, representation alignment (REPA) (Yu et al., 2025), leverages high-quality representations extracted from large-scale pre-trained transformer models, such as DINOv2 (Oquab et al., 2024) and CLIP (Radford et al., 2021). REPA explicitly aligns early-stage diffusion transformer features with clean image features from external encoders, thereby encouraging stronger representational capacity. This alignment enables deeper layers to concentrate on high-frequency content, ultimately improving generative performance. However, training such pre-trained visual encoders demands access to large-scale datasets and involves substantial computational costs.

An alternative approach leverages self-representation learning (Zhu et al., 2024; Jiang et al., 2025; Wang & He, 2025) without external encoders. In this framework, self-representation alignment (SRA) (Jiang et al., 2025) adapts teacher–student discriminative pair structures within diffusion models to enable self-supervised knowledge distillation (Zhu et al., 2024). Using the observation that deeper layers produce richer representations (Xiang et al., 2023; Yu et al., 2025), the teacher is assigned to deeper layers with lower noise, while the student is associated with the earlier layers. This strategy has been shown to enhance training effectiveness without relying on pre-trained encoders. Nevertheless, the approach remains fundamentally constrained by the representational capacity of the diffusion model itself.

To assess the effectiveness of learned representations, prior work commonly employs *linear probing* (Alain & Bengio, 2016). In this evaluation protocol, the trained diffusion model is frozen, and features from specific layers are extracted to train a lightweight linear probe (classifier). The classifier's accuracy serves as a measure of the *degree of linear separability* of the feature space. Linear probing accuracy has been shown to correlate strongly with both the training efficiency and the generative quality of diffusion transformers (Leng et al., 2025; Yu et al., 2025). These findings suggest that a key factor underlying the success of prior approaches is the refinement of feature representations, thereby promoting *linear separability*.

**Our Approach:** This work is motivated by a simple yet fundamental question:

*"Can diffusion models learn highly linearly separable representations that improve training efficiency while producing higher-quality outputs without representation alignment or external encoders?"*

To this end, we introduce **L**inear **SEP**arability (LSEP), a training regularization strategy that incorporates a linear probe into diffusion models to simultaneously improve the separability of early-stage hidden representations and optimize the denoising objective.

In particular, we insert a trainable linear probe into an intermediate layer of the diffusion model, as shown in Fig. 1, similar in spirit to linear probing evaluations, but *without freezing* the model parameters. Unlike prior alignment-based approaches, our method does *not require an external visual encoder or explicit representation alignment*, while still promoting stronger linear separability at the intended depth of the diffusion model. Experiments show that LSEP substantially boosts both the training efficiency and the generative output quality on flow-based transformer SiTs (Ma et al., 2024). Our main contributions are as follows:

- We introduce *Linear SEParability (LSEP)*, a framework that integrates a linear probe (classifier) into generative model training to simultaneously enhance linear separability and the standard denoising objective.
- To enable the two objectives to mutually reinforce each other, we propose novel training techniques for the classification term while keeping the denoising training unchanged: (1) classification-specific conditioning for the linear probe branch, (2) random cropping to enhance patch-level linear separability, and (3) time-dependent weighting of the classification loss.

- We demonstrate that LSEP substantially enhances both training efficiency and generative quality in flow-based transformer architectures, without relying on *an external visual encoder or explicit representation alignment*.
- SiT-XL with LSEP converges to lower FID significantly faster than the baseline SiT-XL, achieving performance matching that of the alignment-based model REPA, as shown in Fig. 1 (b). Moreover, it achieves an FID of 1.44 on $256 \times 256$ ImageNet and FID of 1.66 on $512 \times 512$ ImageNet with classifier-free guidance using the guidance interval, establishing the best performance among models without relying on external encoder architectures.
- Finally, we show that LSEP synergistically combines with alignment-based methods to enhance linear separability of representations through distinct mechanisms, and to further improve both training efficiency and generative performance.

## 2 PRELIMINARIES

### 2.1 TRAINING FLOW-BASED DIFFUSION TRANSFORMER

Flow-based approaches (Lipman et al., 2023) learn a velocity field $\mathbf{v}_\theta(\mathbf{x}_t, t)$ that defines a probability flow ordinary differential equation (PF-ODE), characterizing the deterministic evolution of a data point $\mathbf{x}_t$. SiT (Ma et al., 2024) adopts this framework to model a continuous-time forward process:

$$\mathbf{x}_t = \alpha_t \mathbf{x}_0 + \sigma_t \boldsymbol{\epsilon}, \quad \boldsymbol{\epsilon} \sim \mathcal{N}(\mathbf{0}, \mathbf{I}), \tag{1}$$

where $\alpha_t$ and $\sigma_t$ are time-dependent functions such that $\alpha_t$ decreases and $\sigma_t$ increases with $t \in [0, T]$, satisfying $\alpha_0 = \sigma_T = 1$ and $\alpha_T = \sigma_0 = 0$. The PF-ODE is given by $\frac{d\mathbf{x}_t}{dt} = \mathbf{v}_\theta(\mathbf{x}_t, t)$, where the distribution of the ODE solution at time $t$ matches the marginal distribution of the forward process. To train the velocity model $\mathbf{v}_\theta(\mathbf{x}_t, t)$, the following velocity matching loss is minimized:

$$\mathcal{L}_{\text{velocity}} = \mathbb{E}_{\mathbf{x}_0, \boldsymbol{\epsilon}, t} \left[ \| \mathbf{v}_\theta(\mathbf{x}_t, t) - \dot{\alpha}_t \mathbf{x}_0 - \dot{\sigma}_t \boldsymbol{\epsilon} \|^2 \right], \tag{2}$$

where $\dot{\alpha}_t = \frac{d\alpha_t}{dt}$ and $\dot{\sigma}_t = \frac{d\sigma_t}{dt}$ denote the time derivatives of $\alpha_t$ and $\sigma_t$, respectively.

In practice, to incorporate class guidance via classifier-free guidance (CFG) (Ho & Salimans, 2021), the velocity model $\mathbf{v}_\theta$ is trained conditionally on a label $\mathbf{c} \in \mathcal{C}$, resulting in a model of the form $\mathbf{v}_\theta(\mathbf{x}_t, t \mid \mathbf{c})$ (Peebles & Xie, 2023; Ma et al., 2024; Yu et al., 2025). Here, $\mathcal{C}$ includes both the dataset class labels $\{\mathbf{c}_{\text{class}}\}$ and an unconditional label $\mathbf{c}_\varnothing$, *i.e.* $\mathcal{C} = \{\mathbf{c}_{\text{class}}\} \cup \mathbf{c}_\varnothing$. Both types of conditions are employed during training, where the unconditional label is used with a small probability, $\rho_D \ll 1$.

### 2.2 REPRESENTATION ALIGNMENT FOR GENERATIVE MODELS

To improve training efficiency and generation quality of diffusion transformers, REPA (Yu et al., 2025) aligns a model's hidden-layer representations with large-scale pretrained self-supervised visual encoders such as DINOv2 (Oquab et al., 2024) and CLIP (Radford et al., 2021). Specifically, REPA introduces a regularization term that maximizes patch-wise similarity between features $\mathbf{y}_{\text{clean}}$, extracted from the pretrained external encoder, and $k^{\text{th}}$ hidden layer features $\mathbf{h}_t^k$ of the diffusion transformer encoder at timestep $t$. The representation alignment loss is defined as:

$$\mathcal{L}_{\text{repa}} = -\mathbb{E}_{\mathbf{x}_{\text{clean}}, \epsilon, t} \left[ \frac{1}{N} \sum_{n=1}^{N} \text{sim}_{\text{cos}} \left( \mathbf{y}_{\text{clean}}^n, \text{MLP}(\mathbf{h}_t^{k,n}) \right) \right], \tag{3}$$

where $N$ is the total number of patches, $\epsilon$ is the noise level at timestep $t$, and MLP is a trainable multilayer perceptron that projects $\mathbf{h}_t^{k,n}$ to adaptively align it with the representation space of $\mathbf{y}_{\text{clean}}^n$, while $\text{sim}_{\text{cos}}(\cdot, \cdot)$ denotes the cosine similarity function. This regularization term is incorporated into the original diffusion-based objective in Eq. 2 during training, yielding the combined loss:

$$\mathcal{L}_{\text{REPA}} = \mathcal{L}_{\text{velocity}} + \lambda \cdot \mathcal{L}_{\text{repa}}. \tag{4}$$

Large-scale pretrained encoders (*e.g.*, DINOv2) are optimized not only with image-level but also with patch-level objectives, enabling them to learn more robust representations (Zhou et al., 2022; Oquab et al., 2024). From the perspective of the patch-level manifold space (Hao et al., 2022), REPA leverages patch-wise representation alignment to capture these finer-grained structures, thereby maximizing the effectiveness of alignment with such pretrained encoders.

### 2.3 LINEAR PROBING

Linear probing evaluations (Alain & Bengio, 2016) were originally proposed to analyze deep neural networks by measuring the degree of linear separability of features across layers. In this approach, the model under investigation is frozen. Given an input $\mathbf{x}$, the neural network encoder extracts features $\mathbf{h}^k$ at the $k^{\text{th}}$ layer, analogous to Section 2.2, which are subsequently reduced via global pooling to yield the representation $\mathbf{g}^k$. A linear classifier $f^k$ is then trained on these pooled features to solve the classification task by minimizing the cross-entropy loss:

$$\mathcal{L}_{\text{class}} = -\mathbb{E}_{(\mathbf{x},\mathbf{c})}\big[\,\mathbf{c}_{\text{gt}}^{\top} \log\big(f^k(\mathbf{g}^k)\big)\big], \tag{5}$$

where $\mathbf{c}_{\text{gt}}$ is ground-truth label and $f^k(\mathbf{g}^k) = \text{softmax}(\mathbf{W}\mathbf{g}^k + \mathbf{b})$ with $(\mathbf{W}, \mathbf{b})$ as probe parameters.

Recent studies have demonstrated that both transformer-based and CNN-based diffusion models learn semantically meaningful representations through the denoising task in training (Baranchuk et al., 2022; Chen et al., 2025; Xiang et al., 2023), as evidenced by evaluations of their linear separability via linear probing. Furthermore, alignment-based methods have employed linear probing to quantify the improvements in the linear separability of the target layers' representations, demonstrating how such enhancements correlate with training efficiency (Jiang et al., 2025; Yu et al., 2025).

## 3 LSEP: PROMOTING LINEAR SEPARABILITY OF DENOISING NETWORKS

We introduce a simple yet novel trainable linear probe into an intermediate layer of the diffusion model, in the spirit of linear probing evaluations, but train it *without freezing* its model parameters. We conceptualize our proposed method as a harmonious integration of diffusion and linear classification tasks, designed to enhance both simultaneously, as described in Fig. 1. All intermediate layers, along with a linear probe inserted at a pre-specified depth, are simultaneously trained to enhance linear separability of the intermediate representations. The entire diffusion model leverages these well-separated features to perform effective denoising.

Consequently, our model should be carefully designed to accommodate the coexistence of these two distinct tasks. To this end, we introduce several novel training strategies: (1) design of the linear probe branch, (2) task-specific conditioning of the shared intermediate layers, (3) random cropping for improving patch-level linear separability, and (4) time-dependent weighting scheduling of the linear probing loss. Each subsequent subsection provides details on these components, and their respective results are presented in Section 4.3.

### 3.1 LINEAR CLASSIFIER BUILT ON DIFFUSION MODEL

As illustrated in Fig. 2, our linear classifier consists of a normalization module followed by a linear layer, similar in spirit to prior works (Alain & Bengio, 2016; He et al., 2022; Chen et al., 2025; Yu et al., 2025). Unlike these works, which may or may not employ batch normalization, we adopt layer normalization to ensure stable training in the presence of the denoising objective. Thus, intermediate features extracted from

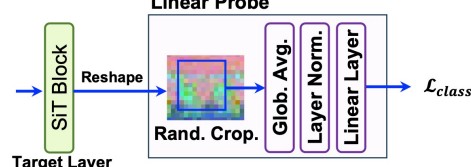

Figure 2: Architectures of the linear classifier.

the SiT model are globally aggregated via average pooling for dimensionality reduction, followed by layer normalization and a linear classification head. The diffusion transformer is trained up to a specified target depth jointly with the classifier. This joint optimization encourages early-stage representations to become more linearly separable, guided by supervision from the classifier. The classification loss is given in Eq. 5, with the only modification being the addition of layer normalization.

Our proposed total loss, which incorporates the scaling factor $\omega_{\text{class}}$ for the classification term to control the trade-off between classification and denoising tasks, together with Eq. 2, is defined as:

$$\mathcal{L}_{\text{LSEP}} = \mathcal{L}_{\text{velocity}} + \omega_{\text{class}} \cdot \mathcal{L}_{\text{class}}. \tag{6}$$

### 3.2 CONDITIONING TO LINEAR PROBE BRANCH

In practice, conditioning of the diffusion transformer consists of time and class embeddings, which are summed and applied through projection operators. For time embedding, we use the same timestep

conditioning for both tasks, which helps the classifier to align with the temporal dynamics of the denoising process. However, we apply a different class conditioning to the linear probe branch, as indicated by the blue path in Fig. 1 to enhance linear separability of intermediate features. This is because the class embedding is a function of the class label, $\mathbf{c}_{\text{class}}$. Thus, this information may cause shortcut learning, where the linear classifier relies directly on class conditioning to perform classification rather than learning linearly separable representations.

To mitigate this, we assign the unconditional class label ($\mathbf{c}_{\varnothing}$) as $\mathbf{c}_{\text{class}}$ with probability $\rho_L$, chosen to be close to 1. This withholds the class information from the classifier and prevents it from relying solely on class conditioning. With probability $1 - \rho_L$, we assign another class label $\mathbf{c} \in \mathcal{C} - \{\mathbf{c}_{\varnothing}\}$ as $\mathbf{c}_{\text{class}}$, which exposes the classifier to non–affine-transformed features. Note this approach is different from the class conditioning used for the denoising task of the diffusion model, where non-null class information is included in the class conditioning with higher probability, $1 - \rho_D$. Our combined strategy encourages feature representations in the early stages to capture meaningful semantics.

### 3.3 Random Cropping for Improving Patch-level Linear Separability

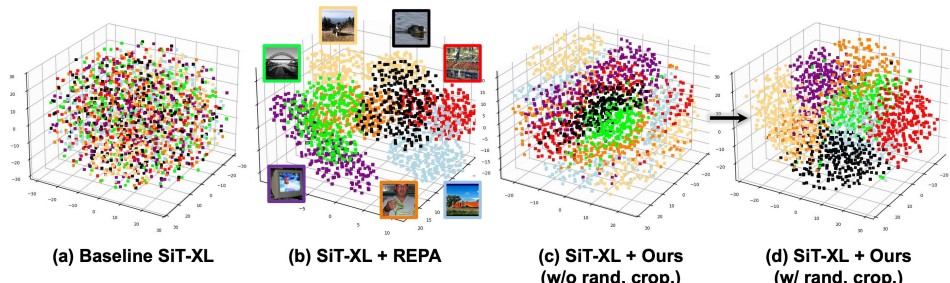

| (a) Baseline SiT-XL | (b) SiT-XL + REPA | (c) SiT-XL + Ours (w/o rand. crop.) | (d) SiT-XL + Ours (w/ rand. crop.) |

Figure 3: t-SNE 3D projection visualization of the patch-level manifold space of SiT-XL using baseline, REPA, and our method with and without random cropping (400K iters, $t$=0.7). Each patch within the $8^{th}$ layer intermediate features is represented by a square, with distinct colors for 7 classes.

Our linear probe uses a summary statistic of the intermediate features, obtained via global pooling, to perform its classification task. While this ensures linear separability of the pooled feature vector, the patch-level features may not be as optimally separated in the patch-level manifold space (Hao et al., 2022). To further investigate this, we performed a t-SNE (Maaten & Hinton, 2008) 3D visualization in the patch-level manifold space, as illustrated in Fig. 3. In Fig. 3 (b), the alignment-based method (REPA) forms well-separated clusters by leveraging patch-wise alignment with an extensively pre-trained external encoder. Our method (Fig. 3 (c)), optimized with globally pooled features, achieves improved clustering compared to the baseline (a). However, certain patches (in light orange and light blue) exhibit a dispersed distribution.

To address this, we randomly crop the intermediate feature map into $n \times n$ patches and compute the mean over this subset before feeding it to the linear classifier as illustrated in Fig. 2. Concretely, we first reshape the intermediate features from $\mathbf{h}^k \in \mathbb{R}^{T \times D}$ to $\mathbb{R}^{t \times t \times D}$, where $T$ is the length of features, $D$ is the channel dimension, and $t^2 = T$. We then randomly crop the features to $\mathbb{R}^{n \times n \times D}$, with $n \leq t$. This strategy enhances the separability of both the mean vectors and subsets of patch-level features, while also introducing diversity similar to data augmentation. As illustrated in Fig. 3 (d), this promotes clearer cluster formation and further contributes to improved denoising training.

### 3.4 Time-Dependent Weighting Scheduling of $\mathcal{L}_{class}$

One of the major differences between standard classification and classification within our designed diffusion model is that the inputs are combined with different noise levels, resulting in a multitude of input distributions. Although the time embedding effectively tracks these variations and guides the corresponding denoising tasks, a single linear probe has limitations in classifying across such diverse distributions. While incorporating a time-dependent embedding into the classifier or using multiple classifiers for different time steps may be viable, our focus is not on enhancing the classifier head. Instead, we maintain a single classifier and introduce diversity through weight scheduling to improve the linear separability of intermediate transformer blocks.

Table 1: Performance comparison of the proposed key training strategies on the SiT-L/2 model. Each strategy is distinguished by a different color, and dark colors indicate the best option within each component. The symbols ↑ and ↓ indicate that higher and lower values are better, respectively.

| Iter. | Uncond. Prob. | Target | Rand. Crop ($n \times n$) | $\omega_{\text{class}}$ | FID↓ | sFID↓ | IS↑ | Pre↑ | Rec↑ |
|---|---|---|---|---|---|---|---|---|---|
| 400K | | | **Baseline SiT-L/2** | | 18.8 | 5.29 | 72.0 | 0.64 | 0.64 |
| 400K | 0.1 | 8 | $n = 16$ | 0.03 | 20.6 | 5.47 | 70.2 | 0.62 | 0.65 |
| | 0.8 | 8 | | | 13.1 | 5.33 | 95.2 | 0.67 | 0.63 |
| | **0.9** | 8 | | | 12.9 | 5.37 | 96.2 | 0.67 | 0.64 |
| | 1.0 | 8 | | | 14.1 | 5.34 | 91.0 | 0.66 | 0.64 |
| 400K | 0.9 | 6 | $n = 16$ | 0.03 | 12.8 | 5.41 | 96.2 | 0.67 | 0.64 |
| | | **7** | | | 12.7 | 5.34 | 98.1 | 0.67 | 0.64 |
| | | 8 | | | 12.9 | 5.37 | 96.2 | 0.67 | 0.64 |
| | | 9 | | | 13.9 | 5.37 | 91.5 | 0.66 | 0.65 |
| | | 10 | | | 13.4 | 5.43 | 93.0 | 0.66 | 0.64 |
| 400K | 0.9 | 7 | $n = 16$ | 0.03 | 12.7 | 5.34 | 98.1 | 0.67 | 0.64 |
| | | 7 | $n \in [11, 16] \cap \mathbb{Z}$ | 0.03 | 13.0 | 5.33 | 95.8 | 0.67 | 0.64 |
| | | 7 | $\mathbf{n \in [12, 16] \cap \mathbb{Z}}$ | 0.03 | 12.5 | 5.34 | 98.8 | 0.67 | 0.64 |
| | | 7 | $n \in [13, 16] \cap \mathbb{Z}$ | 0.03 | 13.2 | 5.33 | 95.2 | 0.67 | 0.64 |
| | | 7 | $n \in [14, 16] \cap \mathbb{Z}$ | 0.03 | 12.5 | 5.37 | 97.6 | 0.68 | 0.64 |
| 400K | 0.9 | 7 | $n = 16$ | 0.03 | 12.7 | 5.34 | 98.1 | 0.67 | 0.64 |
| | | 7 | $n = 16$ | $[0.02, 0.03]_{10 \text{ Bins}}$ | 12.5 | 5.33 | 97.4 | 0.67 | 0.64 |
| | | 7 | $n = 16$ | $[0.02, 0.03]_{5 \text{ Bins}}$ | 12.5 | 5.36 | 98.6 | 0.67 | 0.64 |
| | | 7 | $n \in [12, 16] \cap \mathbb{Z}$ | $[0.02, 0.025]_{10 \text{ Bins}}$ | 12.5 | 5.32 | 97.9 | 0.67 | 0.63 |
| | | 7 | $n \in [12, 16] \cap \mathbb{Z}$ | $[0.0275, 0.0325]_{5 \text{ Bins}}$ | 12.6 | 5.41 | 97.7 | 0.67 | 0.64 |
| | | 7 | $n \in [12, 16] \cap \mathbb{Z}$ | $\mathbf{[0.0275, 0.0325]_{10 \text{ Bins}}}$ | 12.3 | 5.40 | 98.3 | 0.68 | 0.64 |

To this end, we apply a time-dependent piecewise constant weighting scheme to the linear probe loss, allowing the linear probe to assign different weights according to the noise level and thus learn more effectively. This approach divides the different noise level distributions into $k$ groups, allowing the linear probe to be optimized for each group. It also assigns larger weights to higher noise levels, which helps the classifier perform more effectively on noisier inputs. The time-dependent weight, $\omega_{\text{class}}(t)$ with $k$ bins is defined as:

$$\omega_{\text{class}}(t, k) = \omega_{\text{start}} + \lfloor t \cdot k \rfloor \cdot \Delta\omega, \quad t \in [0, 1] \tag{7}$$

where $\Delta\omega = (\omega_{\text{max}} - \omega_{\text{min}})/k$, and $\omega_{\text{min}}$ and $\omega_{\text{max}}$ are the minimum and maximum weighting values, respectively. We denote this as $\omega_{\text{class}}(t, k) = [\omega_{\text{start}}, \omega_{\text{end}}]_{k \text{ bins}}$ for future reference.

## 4 EXPERIMENTAL SETUP

### 4.1 IMPLEMENTATION DETAILS

We closely follow the experimental setup of SiT (Ma et al., 2024) and REPA (Yu et al., 2025), unless stated otherwise. All models are trained and evaluated on ImageNet (Deng et al., 2009), where images are preprocessed to a resolution of $256 \times 256$ following the data preprocessing protocol of ADM (Dhariwal & Nichol, 2021). We employ the Base, Large, and X-Large model variants introduced in SiT (Ma et al., 2024), all configured with a patch size of 2. We insert a linear classifier after the 4[th], 7[th], and 8[th] layers in the SiT-B/2, SiT-L/2, and SiT-XL/2 models, respectively.

To ensure fair comparison with prior work (Ma et al., 2024; Yu et al., 2025), we train all models using a consistent global batch size of 256. Optimization is performed using AdamW (Kinga et al., 2015; Loshchilov & Hutter, 2019) with a constant learning rate of $1 \times 10^{-4}$ for training the diffusion model. We provide the detailed hyperparameters in Appendix A.

### 4.2 EVALUATION PROTOCOL

For evaluating image generation quality, we strictly follow the ADM evaluation protocol (Dhariwal & Nichol, 2021). We report several standard metrics, including Fréchet Inception Distance (FID) (Heusel et al., 2017), Structural FID (sFID) (Nash et al., 2021), Inception Score (IS) (Salimans et al., 2016),

and Precision and Recall (Kynkäänniemi et al., 2019), all computed using 50K generated samples. Following the sampling procedures from SiT (Ma et al., 2024) and REPA (Yu et al., 2025), we adopt the SDE-based Euler–Maruyama sampler with 250 steps. All evaluations are performed on 50K validation images from the ImageNet dataset (Deng et al., 2009), resized to 256×256 resolution.

### 4.3 ANALYSIS OF KEY STRATEGIES

We present detailed experimental results for the SiT-L/2 model in Tab. 1, effectively serving as an ablation study for the training innovations outlined in Section 3.

**Class Conditioning for Linear Probe** The unconditioning probability $\rho_L$ indicates the proportion of the null class $\mathbf{c}_\varnothing$, as in Section 3.2. When the unconditioning probability is set to 0.1, matching that of the denoising model, the linear probe relies on the conditioning label for shortcut learning. This not only fails to improve the linear separability of intermediate features but also degrades generative performance. Conversely, setting it to $\rho_L = 1.0$ (*i.e.*, using only $\mathbf{c}_\varnothing$) prominently improves generative performance. However, this also causes the class conditioning for the classifier and the denoiser to be learned independently, leading to inconsistencies in class representation. Therefore, we find that using $\rho_L = 0.9$ provides the optimal conditioning ratio. Further analysis is provided in Appendix B.1.

**Target Depth** Numerous works have demonstrated that efficient training of diffusion models depends on representation quality in early layers (Yu et al., 2025; Jiang et al., 2025). Our results align with these insights, showing that the optimal depth for incorporating the linear classifier corresponds to the shallow layers of the model. When the depth is shifted toward the middle layers (*e.g.*, 9 or 10 for SiT-L/2), the generative performance degrades, as the reduced capacity for denoising limits overall effectiveness. Conversely, connecting to very early layers also hampers denoising training. We empirically find that layer 7 yields the best results for SiT-L, and adopt this configuration in our experiments. For SiT-B and SiT-XL, the optimal depths are found to be layers 4 and 8, respectively.

**Random Cropping** To enhance linear separability in the patch-level manifold space as shown in Fig. 3, we investigated the effects of random cropping with varying box sizes. We found that selecting a box size randomly from the range 12 to 16 yielded the best results. This introduces variability, providing more diverse feature samples for the linear classifier and improving separability not only of pooled features but also within the patch-level manifold space.

**Time-dependent $\omega_{\text{class}}$** The first three rows compare a time-dependent $\omega_{\text{class}}$ to a constant $\omega_{\text{class}}$ in the absence of random cropping. This demonstrates that assigning different weights to $\mathcal{L}_{\text{class}}$ over time, using a piecewise constant schedule with multiple bins for training a linear classifier improves the generative results. The last three rows further examine different weight intervals and numbers of bins when time-dependent $\omega_{\text{class}}$ is combined with random cropping approach. The results show $[0.0275, 0.0325]_{10\text{ Bins}}$ achieves the best performance, and is used for the remainder of the study.

## 5 RESULTS

### 5.1 LINEAR SEPARABILITY

We evaluate the effect of promoting linear separability using linear probing, 3D projections, and PCA visualization for SiT-XL with our proposed method. Fig. 4 (a) shows that LSEP achieves higher overall linear probing accuracy across layers, particularly at the early stages. Unsurprisingly, linear probing performance is substantially improved using the LSEP training strategy that jointly targets a linear classification task. In Fig. 4 (b), our proposed method produces well-defined clusters, comparable to those observed in REPA, which indicates that the classes are clearly linearly separable, even though no external encoder information is used. We also provide PCA visualizations in Appendix B.3, which demonstrate that LSEP preserves clearly separable components across varying noise levels.

### 5.2 QUANTITATIVE RESULTS

**Analysis Across Model Sizes.** Tab. 2 summarizes generative performance of models of different sizes with different training paradigms. Note the number of parameters for REPA *excludes* the addi-

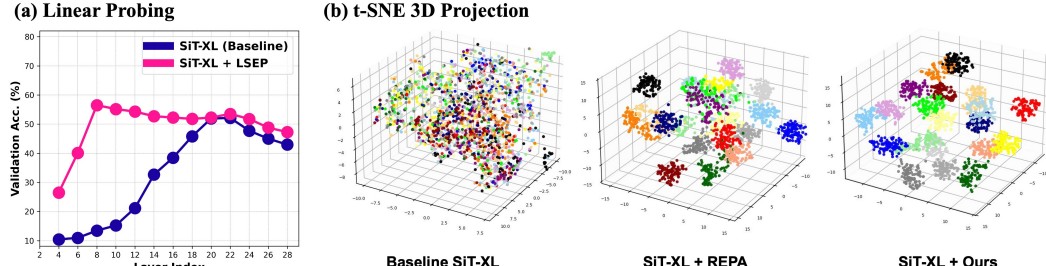

Figure 4: (a) Linear probing evaluation results on baseline SiT-XL and SiT-XL+LSEP with $t = 0.1$. (b) t-SNE 3D projections for baseline SiT-XL (left), alignment-based REPA (middle), and proposed method (right). Intermediate features are extracted from the $8^{th}$ layer of each pre-trained SiT-XL model with $t = 0.1$. Each feature is globally pooled and sampled from 20 randomly selected classes, with 100 samples per class. The representations are then projected into 3D space, with each class visualized using a distinct color. For all methods, the 400K-iteration checkpoint was used.

tional parameters related to the external encoder. Our method consistently improves the generation FID compared to the baseline SiT baseline trained for the same 400K iterations. Notably, FID of 12.3 (Tab. 2) and IS of 98.3 (Tab. 1) achieved by our LSEP (SiT-L) outperform the 12.5 and 90.7, respectively, reported for REPA (SiT-L) (Yu et al., 2025) with pretrained external MAE-L encoder (He et al., 2022), which contains an additional 304M parameters in addition to the transformer diffusion parameters.

While LSEP shows substantial gains in early stages of trainings, we emphasize that LSEP continues to steadily improve FID throughout training, with consistent gains: by 800 epochs, its performance approaches that of REPA, while remaining well beyond the reach of baseline training at 7M iterations, as illustrated in Fig. 1 (b). Furthermore, our approach achieves these promising results *without relying on any pre-trained external models or explicit representation alignment*. Selected qualitative results for SiT-XL/2 using our method are presented in Fig. 5. More qualitative results are provided in Appendix F.

Table 2: FID comparisons on $256 \times 256$ ImageNet, without employing CFG.

| Model | #Params | Epochs | FID↓ |
|---|---|---|---|
| SiT-B/2 | 130M | 80 | 33.0 |
| SiT-B/2 + REPA | 137M | 80 | 24.4 |
| SiT-B/2 + LSEP (ours) | 131M | 80 | 28.3 |
| SiT-B/2 + REPA + **LSEP (ours)** | 139M | 80 | **20.5** |
| SiT-L/2 | 458M | 80 | 18.8 |
| SiT-L/2 + REPA | 466M | 80 | 9.7 |
| SiT-L/2 + LSEP (ours) | 459M | 80 | 12.3 |
| SiT-L/2 + REPA + **LSEP (ours)** | 467M | 80 | **9.5** |
| SiT-XL/2 | 675M | 80 | 17.2 |
| SiT-XL/2 + REPA | 683M | 80 | 7.9 |
| SiT-XL/2 + LSEP (ours) | 676M | 80 | 10.4 |
| SiT-XL/2 + REPA + **LSEP (ours)** | 684M | 80 | **7.5** |
| SiT-XL/2 | 675M | 1400 | 8.3 |
| SiT-XL/2 + REPA | 683M | 800 | 5.9 |
| SiT-XL/2 + LSEP (ours) | 676M | 800 | 5.9 |
| SiT-XL/2 + REPA + **LSEP (ours)** | 684M | 400 | 5.9 |
| SiT-XL/2 + REPA + **LSEP (ours)** | 684M | 800 | **4.7** |

**Combining Alignment-Based Methods with LSEP.** REPA improves the representations by performing token-wise alignment, while our LSEP enhances linear separability by mainly utilizing mean representation vectors. Both methods individually improve linear separability; however, when used together, they are expected to provide more powerful representation geometry and improved training

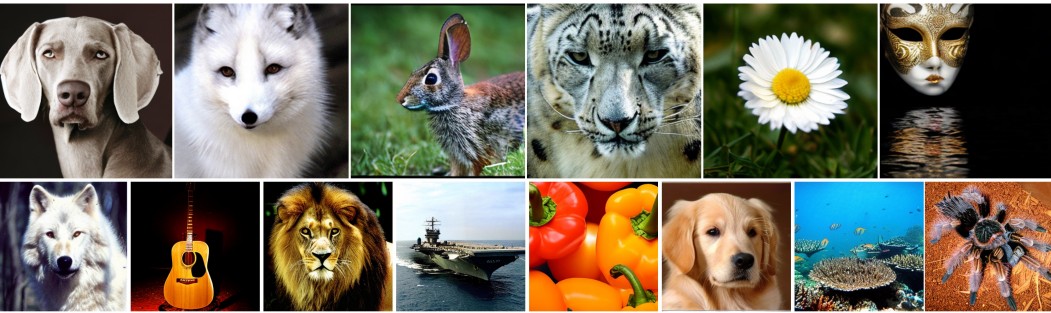

Figure 5: Selected samples on ImageNet $256 \times 256$ from the SiT-XL/2 model with LSEP. We use classifier-free guidance with $\omega_{cfg} = 4.0$.

Table 3: **System-level comparison** on ImageNet $256 \times 256$ with CFG. Within each architecture, the **best** on each metric are bolded. Arrows indicate whether lower ($\downarrow$) or higher ($\uparrow$) values are better.

| Model | Epochs | Tokenizer | FID↓ | sFID↓ | IS↑ | Pre.↑ | Rec.↑ |
|---|---|---|---|---|---|---|---|
| **Pixel diffusion** | | | | | | | |
| ADM-U (Dhariwal & Nichol, 2021) | 400 | - | 3.94 | **6.14** | 186.7 | **0.82** | **0.52** |
| VDM++ (Kingma & Gao, 2023) | 560 | - | **2.40** | - | **225.3** | - | - |
| Simple diffusion (Hoogeboom et al., 2023) | 800 | - | 2.77 | - | 211.8 | - | - |
| CDM (Ho et al., 2022) | 2160 | - | 4.88 | - | 158.7 | - | - |
| **Latent diffusion, U-Net** | | | | | | | |
| LDM-4 (Rombach et al., 2022) | 200 | LDM-VAE | **3.60** | - | **247.7** | **0.87** | **0.48** |
| **Latent diffusion, Transformer *with* pre-trained external encoder** | | | | | | | |
| SiT + REPA (Yu et al., 2025) | 800 | SD-VAE | 1.42 | 4.70 | 305.7 | 0.80 | 0.65 |
| SiT + REPA (Yu et al., 2025) + LSEP (ours) | 800 | SD-VAE | 1.37 | 4.60 | 303.3 | **0.80** | 0.64 |
| LightningDiT (Yao et al., 2025) | 800 | VA-VAE | 1.35 | 4.15 | 295.3 | 0.79 | 0.65 |
| REPA-E (Leng et al., 2025) | 800 | VA-VAE | **1.26** | **4.11** | **314.9** | 0.79 | **0.66** |
| **Latent diffusion, Transformer *without* pre-trained external encoder** | | | | | | | |
| DiT-XL/2 (Peebles & Xie, 2023) | 1400 | SD-VAE | 2.27 | 4.60 | 278.2 | 0.83 | 0.57 |
| SiT-XL/2 (Ma et al., 2024) | 1400 | SD-VAE | 2.06 | **4.50** | 270.3 | 0.82 | 0.59 |
| SD-DiT (Zhu et al., 2024) | 480 | SD-VAE | 3.23 | - | - | - | - |
| MaskDiT (Zheng et al., 2024) | 1600 | SD-VAE | 2.28 | 5.67 | 276.6 | 0.80 | 0.61 |
| DiT + TREAD (Krause et al., 2025) | 740 | SD-VAE | 1.69 | 4.73 | 292.7 | **0.81** | 0.63 |
| SiT + SRA (Jiang et al., 2025) | 800 | SD-VAE | 1.58 | 4.65 | **311.4** | 0.80 | 0.63 |
| SiT + LSEP (ours) | 800 | SD-VAE | **1.44** | 4.73 | 296.8 | 0.80 | **0.64** |

efficiency. As shown in Tab. 5.2, combining REPA with LSEP consistently drives FID even lower, proving that this synergy not only further accelerates training but also enhances generative quality. Additional analysis using patch-level 3D projection visualizations is presented in Appendix B.2.

**System-Level Comparison.** In Tab. 3, we report evaluation results using CFG (Ho & Salimans, 2021) with the guidance interval (Kynkäänniemi et al., 2024). Our method achieves a best FID of 1.44 among models that do not use an external visual encoder and is comparable to alignment-based models, such as REPA, which rely on pretrained external encoders.

# 6 DISCUSSION

**Limitations.** The scope of this paper does not cover advanced generative settings such as text-to-image generation, video generation, and 2K higher-resolution images. Further investigation is warranted to evaluate the applicability and scalability of our framework in these scenarios.

**Future Directions.** Similar to how REPA variants (Yao et al., 2025; Leng et al., 2025) improve generative performance by tuning the VAE through representation alignment, LSEP inherently offers an alternative means to achieve this. Furthermore, since most vision foundation models are transformer-based, prior studies on representation alignment have focused mainly on transformer diffusion models. In contrast, LSEP can also be applied to CNN-based models, broadening its applicability. These ideas will be investigated in future studies.

# 7 CONCLUSION

In this paper, we proposed LSEP, a regularization approach that independently enhances the linear separability of generative models without relying on large-scale pre-trained encoders or representation alignment. We demonstrated that linear separability is a core principle of diffusion training and repurposed linear probing, which is typically used for post-hoc evaluation, as an effective training tool. Our results show that LSEP improves both training efficiency and generative performance, achieving a state-of-the-art FID score of 1.44 with a single model, without the need for any alignment.

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

## A  Implementation Details

Table 4: Hyperparameter setup for different SiT models and LSEP.

| | SiT-B/2 | SiT-L/2 | SiT-XL/2 |
|---|---|---|---|
| **Architecture** | | | |
| Input dim. | $32 \times 32 \times 4$ | $32 \times 32 \times 4$ | $32 \times 32 \times 4$ |
| Num. layers | 12 | 24 | 28 |
| Hidden dim. | 768 | 1,024 | 1,152 |
| Num. heads | 12 | 16 | 16 |
| **LSEP** | | | |
| Uncon. Prob. | 90% | 90% | 90% |
| Target depth | 4 | 7 | 8 |
| Rand. Crop | [14, 16] | [12, 16] | [12, 16] / [24, 32] (Tab. 6) |
| $\omega(t)$ | $[0.005, 0.01]_{10 \text{ bins}}$ | $[0.0275, 0.0325]_{10 \text{ bins}}$ | $[0.0225, 0.03]_{10 \text{ bins}}$ |
| lr for linear probes. | 0.03 | 0.0001 | 0.0001 |
| **Optimization** | | | |
| Training epochs | 80 | 80 | 80 / 800 (Fig. 1) |
| Batch size | 256 | 256 | 256 / 512 (Fig. 1) |
| Optimizer | AdamW | AdamW | AdamW |
| lr | 0.0001 | 0.0001 | 0.0001 |
| $(\beta_1, \beta_2)$ | (0.9,0.999) | (0.9,0.999) | (0.9,0.999) |
| **Interpolants** | | | |
| $\alpha_t$ | $1 - t$ | $1 - t$ | $1 - t$ |
| $\sigma_t$ | $t$ | $t$ | $t$ |
| $w_t$ | $\sigma_t$ | $\sigma_t$ | $\sigma_t$ |
| Training objective | v-prediction | v-prediction | v-prediction |
| Sampler | Euler-Maruyama | Euler-Maruyama | Euler-Maruyama |
| Sampling steps | 250 | 250 | 250 |
| Guidance | - | - | $\omega_{cfg} = 1.7$ (Tab.3) |
| | - | - | Interval : [0, 0.675] (Tab.3) |

Strictly following the experimental setups of SiT (Ma et al., 2024) and REPA (Yu et al., 2025) for the denoising loss, we integrate the linear probe into our model and employ the hyperparameters listed in Tab. 4. We use an increased learning rate for the linear classifier in the SiT-B/2 model, which significantly improves both training efficiency and generative performance. On the other hand, for the Large and X-Large models, this adjustment does not lead to improvements in generative results. Experiments on ImageNet 256 × 256 were conducted using 4 NVIDIA A100 GPUs for the 80-epoch setup and 16 NVIDIA A100 GPUs for the 800-epoch setup. Experiments on ImageNet 512 × 512 were conducted using 32 NVIDIA A100 GPUs.

## B  Additional Analysis

### B.1  Conditioning for linear probe branch

As discussed in Section 4.3, conditioning for linear probing plays an important role in improving both linear separability and generation quality. When the unconditioning probability $\rho_L$ in Section 3.2 is set to 0.1, matching that of the denoising model, the linear probe tend to rely on shortcut learning, as illustrated by the yellow curve in Fig. 6. Moreover, as the unconditioning ratio increases, the classifier's reliance on class information gradually diminishes, leading to a more progressive learning process. However, as mentioned earlier, when the conditioning is fully as-

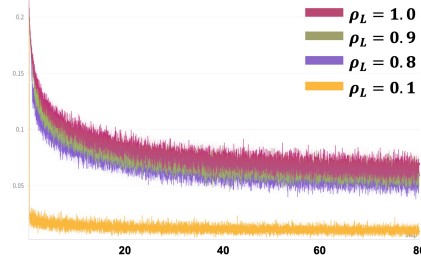

Training Epochs vs. $\omega_{class} \cdot \mathcal{L}_{class}$

Figure 6: Comparison of $\mathcal{L}_{\text{class}}$ across training epochs with varying conditioning ratios in the linear probe branch.

signed to the unconditional class (*i.e.*, $\rho_L = 1.0$), a mismatch arises with the conditioning of the denoising model, which prevents achieving optimal performance. Therefore, incorporating a small proportion of the conditional class $\mathbf{c}_{\text{class}}$ is necessary to obtain the best results.

## B.2 FURTHER ANALYSIS OF COMBINING LSEP AND REPA

**t-SNE 3D Projection (Patch-level)**

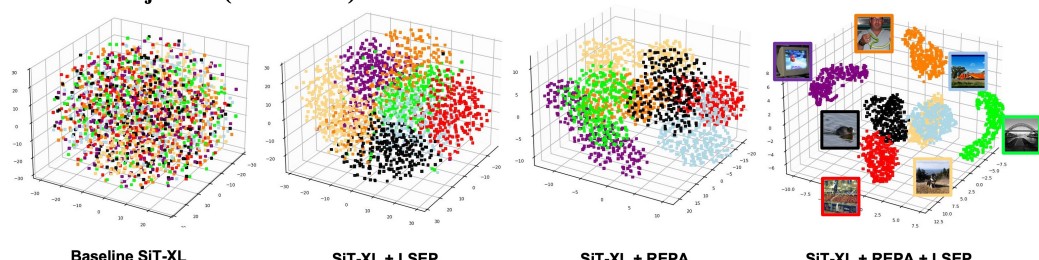

Figure 7: t-SNE 3D projection visualization of baseline SiT-XL, SiT-XL with our method, SiT-XL with REPA, and SiT-XL with REPA combined with our method in the patch-level manifold space (400K iterations, $t = 0.7$), using the same settings as in Fig. 3.

In Section 5.2, we showed that combining LSEP with the alignment-based method (REPA) further improves training efficiency and generative performance of the latter. The patch-level linear separability provided by REPA, together with the mean-vector linear separability from LSEP, enhances the learned representations and thereby strengthens training. As shown in Fig. 7, in t-SNE 3D projection, the combination of REPA and LSEP results in the strongest class-wise separability.

## B.3 PCA VISUALIZATION

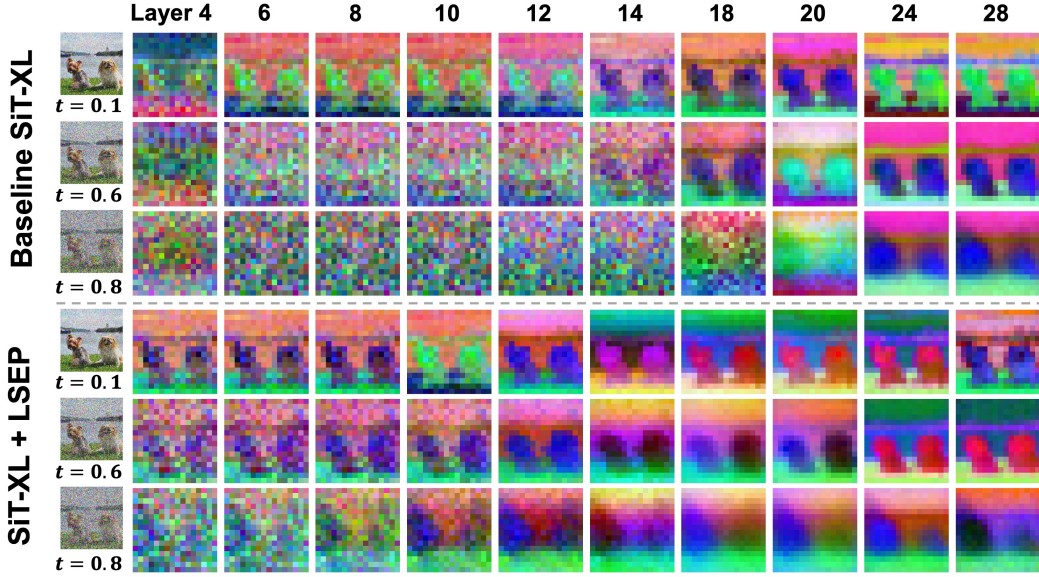

Figure 8: PCA visualizations at various noise levels and layers within the SiT-XL models. Checkpoints at 400K iterations of baseline SiT-XL and SiT-XL + LSEP are used for the visualizations.

We plot the PCA visualization of intermediate features. These demonstrate that LSEP preserves distinctly separable components across different noise levels., with particularly clear separation in the early-stage layers and under higher noise conditions (*e.g.*, layers up to 14, $t = 0.6$ and 0.8).

### B.4 THE STABILITY OF LSEP UNDER PERTURBED (OR NOISY) CLASS-CONDITIONING

LSEP enhances linear separability through conditioning signals (e.g., class labels in the ImageNet case), so it is important and meaningful to evaluate its robustness under perturbed conditioning. To this end, we conducted experiments by intentionally injecting label noise. Specifically, we randomly corrupted 5% and 10% of the 1.28M ImageNet labels and trained SiT-L/2 models using the same hyperparameters as in Tab. 4.

Table 5: Evaluation of LSEP under perturbed class-conditioning.

| Method | Noisy Level | FID | Linear Probing Acc. |
|---|---|---|---|
| SiT-L/2 (Baseline) | 0% | 18.5 | 18.63 |
| SiT-L/2 + LSEP | 0% | 12.3 | 57.43 |
| SiT-L/2 + LSEP | 5% | 14.4 | 55.66 |
| SiT-L/2 + LSEP | 10% | 16.3 | 56.23 |

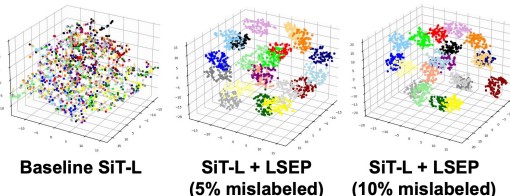

**Baseline SiT-L**  **SiT-L + LSEP (5% mislabeled)**  **SiT-L + LSEP (10% mislabeled)**

Figure 9: 3D t-SNE projection of the results from training SiT-L/2 with mislabeled data at different mismatch levels. Feature extraction follows the same setup as in Fig. 4 at 80 epochs and $t = 0.5$.

As shown in Tab. 5, the FID increases as the proportion of mismatched labels grows, which is expected since noisy supervision degrades the formation of clean linear separability. However, LSEP still consistently outperforms the baseline even under 10% label noise. Importantly, the linear probing accuracy exhibits only a slight degradation. This suggests that while noisy labels introduce a weaker separability signal, LSEP mainly strengthens the intrinsic visual separability and thus remains robust under label noise.

### B.5 LINEAR PROBING EVALUATION ON DIFFERENT LSEP CONFIGURATIONS

Table 6: Linear probing evaluation on different configurations of SiT-L/2 models

| $\rho_L$ | Target | Rand. Crop. | $\omega_{\text{class}}$ | FID↓ | sFID↓ | IS↑ | Pre↑ | Rec↑ | Acc.↑ |
|---|---|---|---|---|---|---|---|---|---|
| | | Baseline SiT-L/2 (Evaluated depth = 8) | | 18.5 | 5.20 | 72.2 | 0.64 | 0.63 | 18.63 |
| 0.9 | 8 | $n = 16$ | 0.03 | 12.9 | 5.37 | 96.2 | 0.67 | 0.64 | 58.05 |
| 0.9 | 7 | $n = 16$ | 0.03 | 12.7 | 5.34 | 98.1 | 0.67 | 0.64 | 56.62 |
| 0.9 | 7 | $n \in [12, 16] \cap \mathbb{Z}$ | 0.03 | 12.5 | 5.34 | 98.8 | 0.67 | 0.64 | 56.87 |
| 0.9 | 7 | $n \in [12, 16] \cap \mathbb{Z}$ | $[0.0275, 0.0325]_{10 \text{ Bins}}$ | 12.3 | 5.40 | 98.3 | 0.68 | 0.64 | 57.43 |

We implemented linear probing tests for each best-performing configuration (highlighted in dark color) within each component in Tab. 1. Linear probing was performed on SiT-L/2 models (80 epochs) using each configuration at $t = 0.1$. For feature extraction depth, we used 8 for the baseline and the selected target depth for the other variants. All experiments followed the same setup: a linear probe was trained on the ImageNet 1.28M training set for 90 epochs with a batch size of 8192, and evaluated on the 50K validation set.

As shown in Tab. 6, all methods improved by LSEP achieve higher linear probing accuracy than the baseline. Although the second row yields the highest linear probing accuracy, we note that accuracy generally increases as the target depth becomes deeper, since more layers contribute to linear separability. Under the same conditions (i.e., using the same target depth), accuracy consistently improves as each designed component is incrementally applied. This supports the observation that linear probing accuracy positively correlates with improvements in generation quality.

Table 7: Number of iterations processed per second for different models using different methods.

| Model | SiT (Baseline) | REPA | LSEP |
|---|---|---|---|
| B/2 | 8.07 iter./s | 7.11 iter./s | 7.75 iter./s |
| L/2 | 4.15 iter./s | 4.01 iter./s | 3.72 iter./s |
| XL/2 | 3.08 iter./s | 2.98 iter./s | 2.67 iter./s |

### B.6 ANALYSIS OF COMPUTATIONAL SPEED

Using 8 A100 GPUs with a global batch size of 256, the number of iterations processed per second is reported in Tab. 7. Although there is a 4 to 13.4 percent decrease in speed compared to the baseline, LSEP achieves a FID of 7.54 at 160 epochs, outperforming the baseline's 8.3 FID at 1400 epochs, which demonstrates that the overall training efficiency remains high. Additionally, since the linear probe branch is not used during inference, it does not affect the inference time.

### B.7 ABLATION STUDY ON $\omega_{\text{class}}$

When designing $\omega_{\text{class}}$, we observed that the primary denoising loss, $\mathcal{L}_{\text{velocity}}$, typically ranges from approximately 0.7 to 0.8 after a few iterations. In contrast, the regularization loss, $\mathcal{L}_{\text{class}}$, starts around 5 to 7 and gradually decreases, slowing down around 1.5 to 3. Since $\mathcal{L}_{\text{class}}$ serves as a regularization term, we selected its weight so that it contributes roughly 10% of the denoising loss, corresponding to a target range of 0.07 to 0.08. Based on this, we chose $\omega_{\text{class}}$ within the range $[0.02, 0.05]$.

Table 8: FID results for different values of $\omega_{\text{class}}$.

| $\omega_{\text{class}}$ | 0.02 | 0.03 | 0.05 |
|---|---|---|---|
| FID | 13.3 | 12.7 | 15.1 |

To validate this choice, we conducted an ablation study analyzing FID variations across different values of $\omega_{\text{class}}$ on the large model. We used $\rho_L = 0.9$ and set the target depth to 7, and we did not apply random cropping to isolate the effect of $\omega_{\text{class}}$. This study confirmed that 0.03 is the most suitable valuein Tab. 8. Building on this optimal value, we then empirically determined the time-dependent weighting schedule in the range $[0.0275, 0.0325]$.

### B.8 DIFFERENCES BETWEEN THE CLASSIFIER IN LSEP AND SUPERVISED LEARNING MODELS

The objectives and training strategies of standard supervised learning (i.e., classification) and the classification component in LSEP are fundamentally different. LSEP is not designed to optimize the classification task itself rather the classification objective functions as a lightweight regularizer that enhances linear separability while jointly training the denoising objective. To promote linear separability rather than classification performance, LSEP purposely uses a *linear* classifier without nonlinearities, whereas traditional supervised learning networks employ deep nonlinear classifiers. Although adding nonlinear heads could increase accuracy, it would contradict our design goal of using the probe to strictly asses linear separability of intermediate features as a regularizer rather than as a high-capacity classifier.

Several structural differences further explain why LSEP exhibits lower classification accuracy than fully supervised models such as ViT, while remaining more suitable for generative training. (1) LSEP applies the linear probe only to a small subset of intermediate layers (up to 8 layers in SiT-XL/2), whereas models like ViT utilize all 24 (or more) transformer layers. (2) These intermediate layers in LSEP are simultaneously used for the denoising task, which constrains their capacity to specialize for classification. (3) LSEP does not employ data augmentation, which further contributes to the lower classification accuracy compared to supervised pipelines.

## C TEXT CONDITIONING IN DIFFUSION TRANSFORMER TRAINING WITH LSEP.

This paper focused on class-conditional generation, where the concept of "linear separability" naturally makes sense among different classes. We use these conditions to enhance the linear separability of intermediate representations. In the text-to-image (T2I) case, the conditioning shifts from class labels to text features extracted from a text encoder such as CLIP. In this scenario, "linear separability" is hard to define since we cannot naturally group text prompts into sets that can be separated. Nonetheless, this raises an important question, and a potential expansion: Can we use a linear combination of the intermediate features to perform different tasks? For a T2I model, an appropriate evaluation target would be similarity with text embeddings.

To this end, we conducted additional experiments during the rebuttal period to assess linear separability in text-based conditioning *without relying on class labels*. As shown in Fig. 10, we converted ImageNet class labels into textual prompts of the form "A photo of [class]" and generated text-

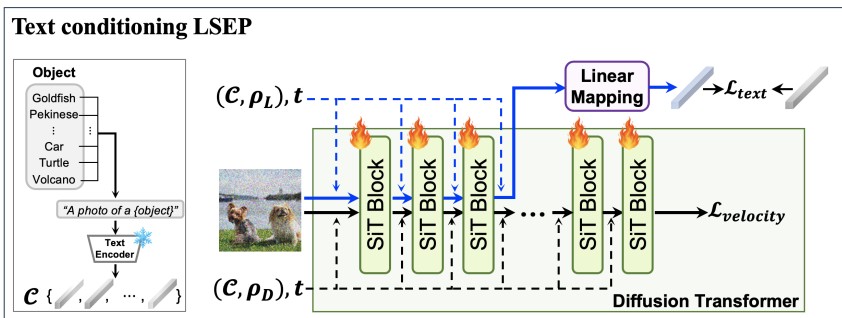

Figure 10: An illustration of text conditioning in diffusion transformer training with LSEP.

conditioned features using CLIP (Radford et al., 2021). The classifier head was removed, and a linear mapping-based text similarity head was introduced. This setup allows us to assess linear separability of intermediate features without actually training a classifier/linear probe, since objects remain distinguishable by class. Note in this case, our auxiliary task for assessing linear separability has shifted from a classification task on categorical variables to a similarity calculation task with continuous variables. LSEP can readily be applied in this setup with the appropriate changes.In practice, the text-conditioning model follows the same conditioning exposure strategy used in class-conditional training (Bao et al., 2023; Yu et al., 2025) utilizing conditional text features with high probability and unconditional text features with low probability for the purpose of CFG.

Table 9: Results on text conditioning training with LSEP

| Method | Conditioning | FID |
|---|---|---|
| SiT-L/2 (Baseline) | Class | 18.8 |
| SiT-L/2 + LSEP | Class | 12.3 |
| SiT-L/2 + LSEP | Text | 15.8 |

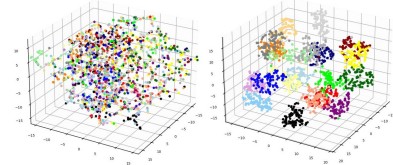

**Baseline SiT-L**  **SiT-L + LSEP (Text Cond.)**

Figure 11: 3D t-SNE projection of the results from training SiT-L/2 with text conditioning. Feature extraction follows the same setup as in Fig. 4 at $t = 0.1$.

As shown in Tab. 9, LSEP with text conditioning as defined in our on SiT-L/2 also improves the FID compared to the baseline. We note that we used the hyperparameters tuned for the classification head rather than those optimized for the new linear mapping–based text similarity. Consequently, the results under text conditioning are not directly comparable to those obtained with class conditioning. Incorporating text prompts to enhance linear separability, as illustrated in the 3D projection example in Fig. 11, further improves model training. This demonstrates that LSEP is not inherently tied to classification tasks and can be extended to broader tasks that operate on linear combinations of the network's intermediate features.

# D EXPERIMENTS ON IMAGENET AT A RESOLUTION OF $512 \times 512$.

We validate the scalability of LSEP by conducting experiments on ImageNet at a resolution of $512 \times 512$. SiT-XL/2 model was used with the same configuration as in Tab. 4, only except that we adjusted the random cropping range to $n \in [24, 32] \cap \mathbb{Z}$ to match the increased feature size of $32 \times 32$ and a batch size of 512.

We trained SiT-XL/2 with LSEP from scratch and followed the fine-tuning strategy used in DDT (Wang et al., 2025), starting from a mid-training checkpoint at a resolution of $256 \times 256$. Specifically, we used the 400 epoch checkpoint of the SiT-XL model trained at $256 \times 256$ and fine-tuned it for an additional 100 epochs.

As shown in Tab. 10, LSEP outperforms REPA even without relying on a large-scale external visual encoder in both the from-scratch and fine-tuning settings, and achieves state-of-the-art performance as a single model at $512 \times 512$ resolution. This demonstrates the strong scalability of LSEP. Representative qualitative results are provided in Fig. 12.

Table 10: **System-level comparison** on ImageNet $512 \times 512$ with CFG. F.T. denotes fine-tuning from models pretrained at $256 \times 256$ resolution. We use classifier-free guidance with $\omega_{\text{cfg}} = 1.9$ and an interval of $[0, 0.75]$.

| Model | Epochs | FID↓ | sFID↓ | IS↑ | Pre.↑ | Rec.↑ |
|---|---|---|---|---|---|---|
| **Pixel diffusion** | | | | | | |
| ADM-G, ADM-U (Dhariwal & Nichol, 2021) | 400 | 3.85 | **5.86** | 221.7 | **0.84** | 0.53 |
| VDM++ (Kingma & Gao, 2023) | - | **2.65** | - | **278.1** | - | - |
| Simple diffusion (U-Net) (Hoogeboom et al., 2023) | 800 | 4.28 | - | 171.0 | - | - |
| **Latent diffusion, Transformer *with* pre-trained external encoder** | | | | | | |
| SiT + REPA (Yu et al., 2025) | 200 | 2.08 | **4.19** | 274.6 | **0.83** | 0.58 |
| DDT (Wang et al., 2025) | 100[F.T] | **1.28** | 4.22 | **305.1** | 0.80 | **0.63** |
| **Latent diffusion, Transformer *without* pre-trained external encoder** | | | | | | |
| DiT-XL/2 (Peebles & Xie, 2023) | 600 | 3.04 | 5.02 | 240.8 | 0.84 | 0.54 |
| SiT-XL/2 (Ma et al., 2024) | 600 | 2.62 | 4.18 | 252.2 | 0.84 | 0.57 |
| MaskDiT (Zheng et al., 2024) | 800 | 2.50 | 5.10 | 256.3 | **0.84** | 0.57 |
| SiT + LSEP (ours) | 200 | 2.10 | **4.18** | 259.0 | 0.83 | 0.57 |
| SiT + LSEP (ours) | 240 | 2.00 | 4.19 | 265.3 | 0.83 | 0.59 |
| SiT + LSEP (ours) | 100[F.T] | **1.66** | 4.56 | **296.8** | 0.81 | **0.61** |

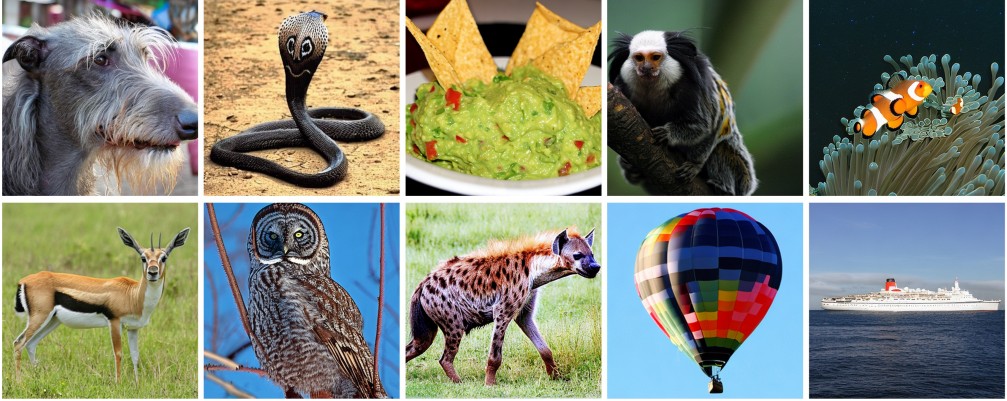

Figure 12: Selected samples on ImageNet $512 \times 512$ from the SiT-XL/2 model with LSEP. We use classifier-free guidance with $\omega_{cfg} = 4.0$.

# E    EXPERIMENTS ON CIFAR-100.

We implemented the LSEP method on the CIFAR-100 dataset (Krizhevsky et al., 2009) using SiT-L/2 to validate its performance on a different dataset and to compare the effects between superclass and fine-grained classes. Specifically, we trained SiT-L/2 models for 4,000 epochs under four settings: (i) SiT-L/2 with fine-grained 100 classes, (ii) SiT-L/2 with 20 super-classes, (iii) SiT-L/2 + LSEP with 100 classes, and (iv) SiT-L/2 + LSEP with 20 super-classes. For LSEP, we used the following configurations: target depth of 7, $\rho_L = 0.9$, random crop: $n \in [12, 16] \cap \mathbb{Z}$, and $\omega_{\text{class}} = [0.005, 0.006]_{10 \text{ steps}}$. This allowed us to investigate how the choice of conditioning clusters affects the training effects.

As shown in Tab. 11, fine-grained 100-class conditioning outperforms training with 20 super-classes for both the baseline SiT-L/2 and SiT-L/2 + LSEP. Notably, for both class settings, adding LSEP results in approximately a twofold speedup in training. These results demonstrate that training efficiency improves even when using superclasses, and that more detailed linear separability further enhances training performance, as one would expect. Overall, this provides clear evidence that improving linear separability has a direct and positive impact on the model's learning efficiency. We provide uncurated class-conditional samples in Fig. 13.

Table 11: Experiments on CIFAR-100 datasets.

| Method | Num. of Class | Epoch | FID |
|---|---|---|---|
| SiT-L/2 | 20 (Superclass) | 4000 | 24.42 |
| SiT-L/2 + LSEP | 20 (Superclass) | 2000 | 16.66 |
| SiT-L/2 + LSEP | 20 (Superclass) | 4000 | **8.08** |
| SiT-L/2 | 100 (Fine-grained) | 4000 | 4.65 |
| SiT-L/2 + LSEP | 100 (Fine-grained) | 2000 | 4.79 |
| SiT-L/2 + LSEP | 100 (Fine-grained) | 4000 | **2.60** |

Another important point that these CIFAR-100 experiments highlight is the critical dependence of REPA on the input size, since it aligns features extracted from an external encoder. For example, to obtain representations from the visual encoder DINOv2, REPA resizes the input to $224 \times 224$, as the model is trained with $224 \times 224$ inputs and a patch size of 14, which maps the input into $16 \times 16$ tokens. As a result, it can be aligned with a SiT model for $256 \times 256$ ImageNet using a patch size of 2 in the latent space. However, for other resolutions such as $512 \times 512$ ImageNet, the input to the visual encoder must be resized to $448 \times 448$, and the positional embeddings must also be interpolated. In other words, additional preprocessing is required to extract representations. Moreover, for low-resolution inputs such as $32 \times 32$ (e.g., CIFAR-100) or $64 \times 64$ (e.g., 64-resolution ImageNet), the visual encoder must be additionally trained, and thus REPA cannot be directly applied. In contrast, LSEP is completely independent of the input size and imposes no such constraints.

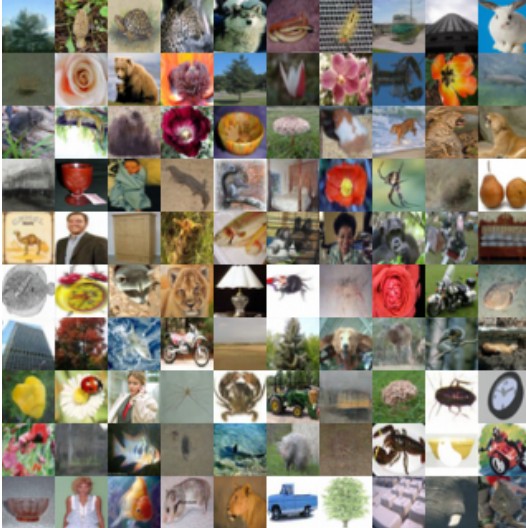

Figure 13: Uncurated class-conditional samples without CFG on CIFAR-100 generated by the SiT-L/2 model with LSEP on fine-grained classes.

## F ADDITIONAL QUALITATIVE RESULTS

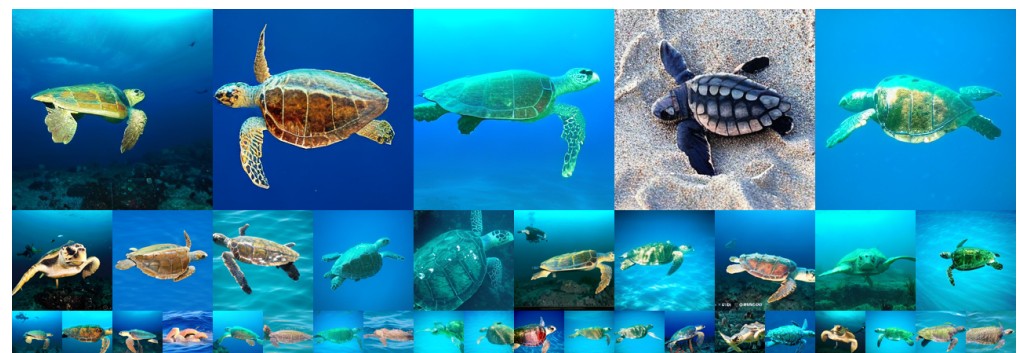

Figure 14: Uncurated samples from SiT-XL/2 + LSEP ($\omega_{\text{CFG}} = 4.0$, class = loggerhead turtle (33))

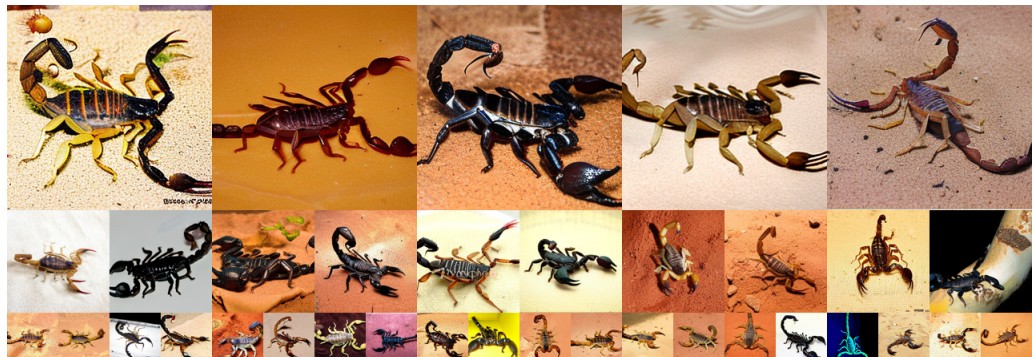

Figure 15: Uncurated samples from SiT-XL/2 + LSEP ($\omega_{\text{CFG}} = 4.0$, class = scorpion (71))

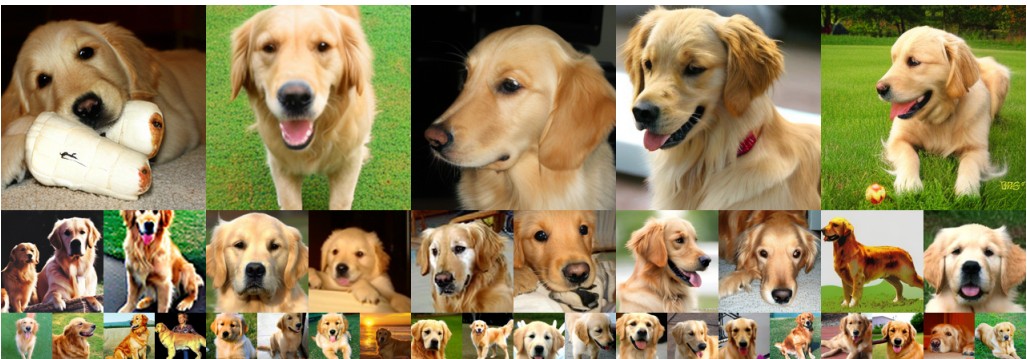

Figure 16: Uncurated samples from SiT-XL/2 + LSEP ($\omega_{\text{CFG}} = 4.0$, class = golden retriever (207))

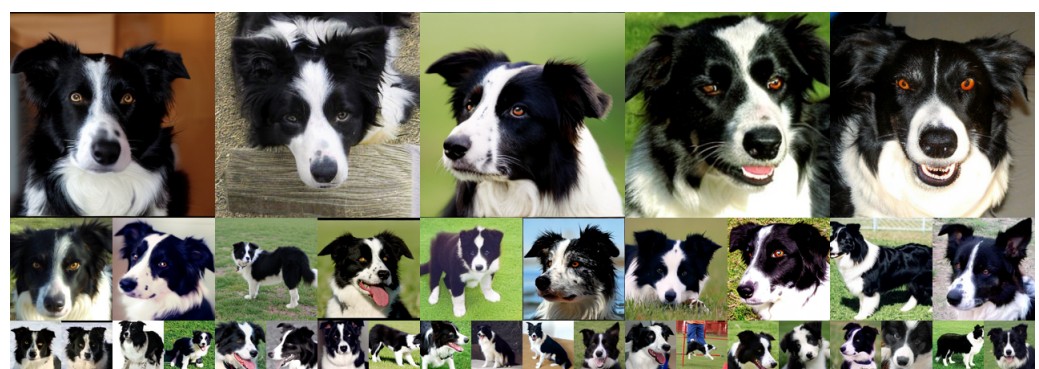

Figure 17: Uncurated samples from of SiT-XL/2 + LSEP ($\omega_{\text{CFG}} = 4.0$, class = border collie(232))

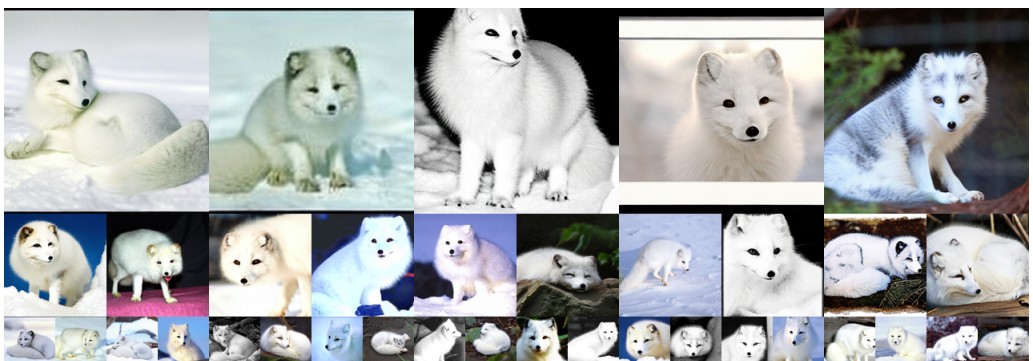

Figure 18: Uncurated samples from SiT-XL/2 + LSEP ($\omega_{\text{CFG}} = 4.0$, class = arctic fox (279))

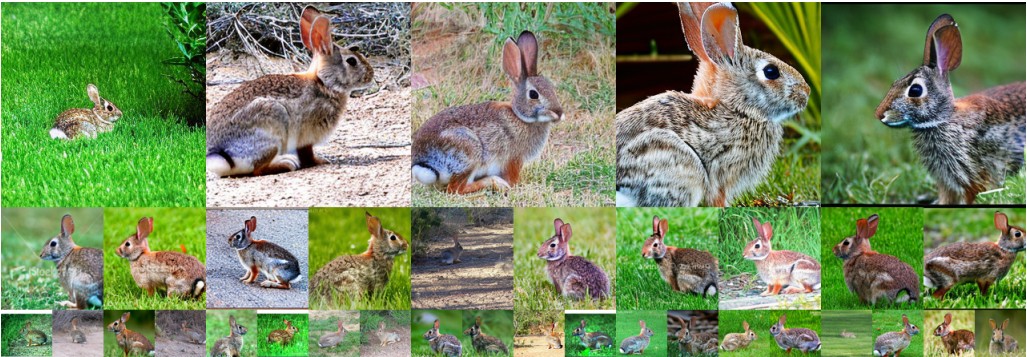

Figure 19: Uncurated samples from SiT-XL/2 + LSEP ($\omega_{\text{CFG}} = 4.0$, class = wood rabbit (330))

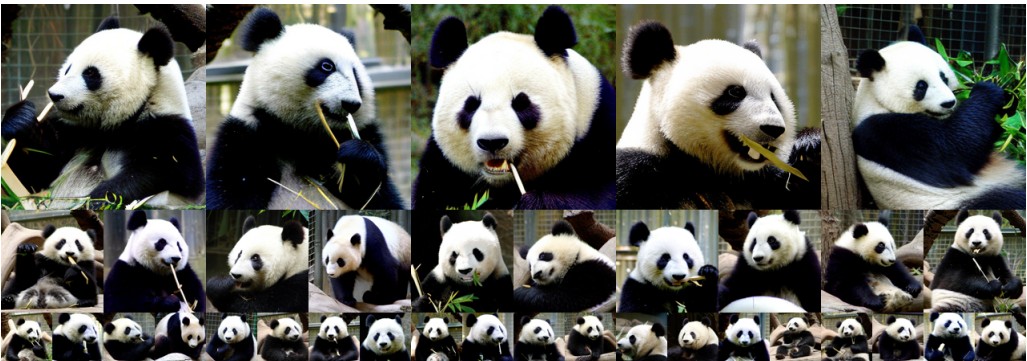

Figure 20: Uncurated samples from SiT-XL/2 + LSEP ($\omega_{\text{CFG}} = 4.0$, class = panda (388))

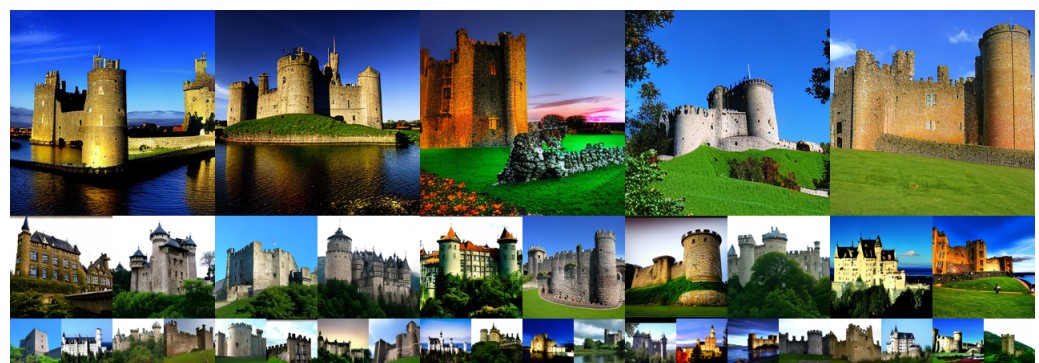

Figure 21: Uncurated samples from SiT-XL/2 + LSEP ($\omega_{\text{CFG}} = 4.0$, class = castle (483))

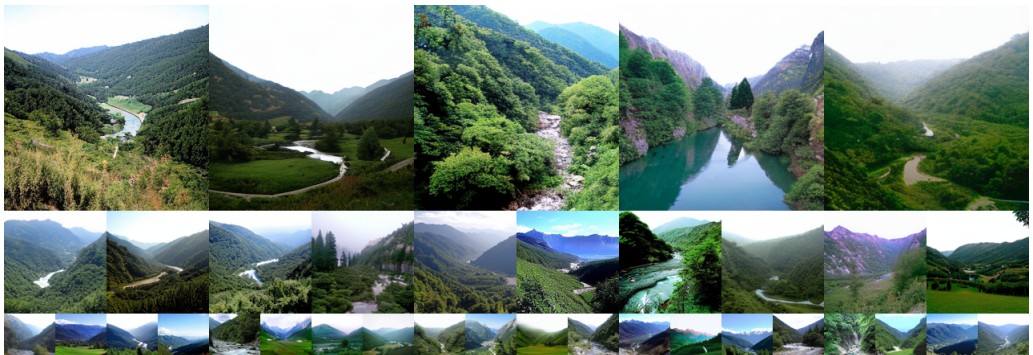

Figure 22: Uncurated samples from SiT-XL/2 + LSEP ($\omega_{\text{CFG}} = 4.0$, class = valley (797))

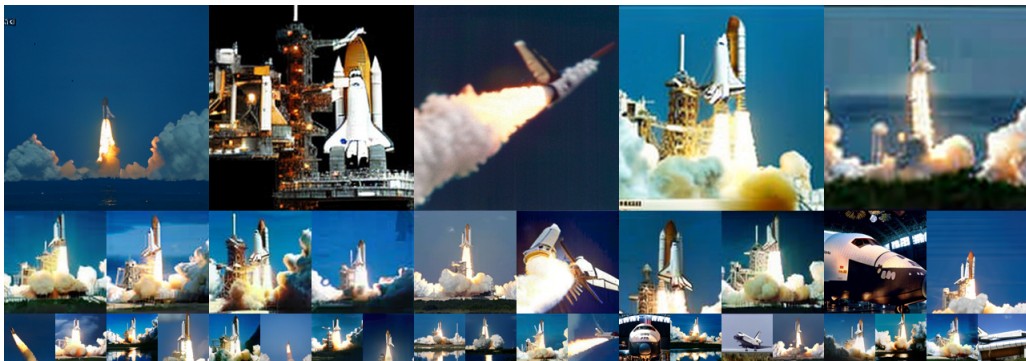

Figure 23: Uncurated samples from SiT-XL/2 + LSEP ($\omega_{\text{CFG}} = 4.0$, class = space shuttle (812))

