# OpenReview forum: "No Alignment Needed for Generation: Learning Linearly Separable Representations in Diffusion Models"
_ICLR.cc/2026/Conference — Submitted to ICLR 2026_

### Official Review · Reviewer_bpbB · 2025-10-30

**Soundness:** 3
**Presentation:** 3
**Contribution:** 2
**Rating:** 4
**Confidence:** 4

**Summary:**

This paper presents Linear Separability (LSEP), a regularization strategy for diffusion transformers which explicitly encourages linearly separable representations without external alignment. The high-level idea is to jointly train a linear probe with the denoising objective thereby enhancing feature separability in the early layer. The authors introduce three design components (1) classification specific conditioning, (2) random cropping for patch-level separability, and (3) time-dependent loss weighting. Experiments on SiT models show that LSEP improves both training speed and generation quality achieving comparable results to alignment-based methods.

**Strengths:**

- The proposed training framework is simple yet efficient, achieving both faster convergence and improved generation quality without relying on external modules.

- The paper is well-written and easy to follow. The motivation and ideas are clearly articulated, figures and visualizations make the concepts easy to understand.

- The ablation studies on each design components are well-designed and detailed which effectively validates each design choice.

**Weaknesses:**

- The study mainly focus on linear probing as the main indicator of representational quality. It is unclear whether this approach generalizes beyond class-labeled settings (e.g., text-to-image or video generation). It would be helpful to discuss how linear separability could be defined or leveraged in such contexts.

- Although linear separability is a useful cue, good linear separation does not necessarily imply better representation. The setup therefore feels somewhat constrained and overly tied to the classification context. When combined with REPA (Table 2 and Figure 7), it achieves much higher performance and clearer separability, showing that linear probing alone does not fully explain the improvements in generation quality.

- The rationale for selecting the weighting parameter $\omega_{\text{class}}$ is not well explained. Its value (e.g., 0.03) appears empirically chosen; an ablation study or sensitivity analysis would strengthen this aspect.

- Higher resolution (e.g., 512x512) experiments would help demonstrate scalability.

**Questions:**

- The linear probe module is already conditioned on timestep embeddings. Could the authors elaborate on why additional time-dependent weight is still necessary? While Table 1 shows empirical improvements, a deeper explanation of its role would be helpful.

- The authors mention that LSEP continues to improve and approaches REPA. Does this trend persist with longer training (e.g., beyond 4M steps)?

-  All experiments are conducted only on SiTs. Is this also applicable to DiTs?

---

> ### Author Response · Authors · 2025-11-21
>
> We thank you for your thoughtful feedback. We are glad that you found our training framework both simple and efficient, and that you appreciated our clear presentation as well as the thorough ablation studies validating each design choice.
> \
> \
> We would like to share several important updates based on additional analyses relevant to all reviewers. Please refer to the “Shared Update’’ section before reading our individual responses.
> \
> \
> **(W1) Generalization of linear separability on the different conditions.**
>
> We thank the reviewer for this great question about a possible extension. This paper focused on class-conditional generation, where the concept of "linear separability" naturally makes sense among different classes. We use these conditions to enhance the linear separability of intermediate representations. In the text-to-image (T2I) case, the conditioning shifts from class labels to text features extracted from a text encoder such as CLIP. In this scenario, "linear separability" is hard to define since we cannot naturally group text prompts into sets that can be separated. Nonetheless, this raises an important question, and a potential expansion: Can we use a linear combination of the intermediate features to perform different tasks? For a T2I model, an appropriate evaluation target would be similarity
> with text embeddings.
>
> To this end, we conducted additional experiments during the rebuttal period to assess linear separability in text-based conditioning *without relying on class labels*. We converted ImageNet class labels into textual prompts of the form “A photo of [class]” and generated text-conditioned features using CLIP. The classifier head was removed, and a linear mapping-based text similarity head was introduced. This setup allows us to assess linear separability of intermediate features without actually training a classifier/linear probe, since objects remain distinguishable by class. Note in this case, our auxiliary task for assessing linear separability has shifted from a classification task on categorical variables to a similarity calculation task with continuous variables. LSEP can readily be applied in this setup with the appropriate changes.In practice, the text-conditioning model follows the same conditioning exposure strategy used in class-conditional training (arxiv:2209.12152, arXiv:2410.06940) utilizing conditional text features with high probability and unconditional text features with low probability for the purpose of CFG.
> \
> \
> *Table. Results on text conditioning training with LSEP*
> |**Method**|**Conditioning**|**FID**|
> |:--------:|:--------------:|:-----:|
> |**SiT-L/2(Baseline)**|Class|18.8|
> |**SiT-L/2+LSEP**|Class|12.3|
> |**SiT-L/2+LSEP**|Text|15.8|
>
> \
> As shown in the table above, LSEP with text conditioning as defined in our on SiT-L/2 also improves the FID compared to the baseline. We note that we used the hyperparameters tuned for the classification head rather than those optimized for the new linear mapping–based text similarity. Consequently, the results under text conditioning are not directly comparable to those obtained with class conditioning. Incorporating text prompts to enhance linear separability further improves model training, demonstrating that LSEP is not inherently tied to a classification task and can be extended to broader tasks that operate on a linear combination of the network's intermediate features. The detailed experimental setup and 3D visualizations are provided in Appendix C of the revised manuscript.
>
> While we were unable to run full T2I experiments on large-scale models such as MMDiT, which would require a new study design, we consider this a promising direction for future research. We thank the reviewer again for suggesting this possible generalization of our idea.

---

> ### Author Response · Authors · 2025-11-21
>
> **(W2) Limitations of LSEP in improving image generation quality alone.**
>
> Thank you for raising this point. In Tab. 2 of the initial submission, LSEP results beyond 80 epochs were not explicitly reported, which made it appear that LSEP alone does not perform well. The revised version incorporates the improved results mentioned in the "Shared Update" in Fig. 1 and Tab. 2.
>
> As in the previous answer (W1), our auxiliary task for assessing linear separability can be shifted from a classification task on categorical variables to a similarity calculation task with continuous variables. This indicates that LSEP is not restricted to the classification context.
>
> While we do agree that REPA and LSEP together achieves exceptional patch-level separability, we note that LSEP alone also achieves good separability, similar to that of REPA (in Figure 7). This similarity is also apparent in their analogous performance outlined in our Shared Update. The additional benefit of using LSEP and REPA together stems from how they individually improve linear separability in different ways. REPA enhances representations by performing token-wise alignment using an external pretrained large-scale visual encoder, which primarily promotes patch-level representation quality. In contrast, LSEP improves linear separability by operating mainly on mean representation vectors. This suggests that combining the two methods should yield a more powerful representation geometry. As shown in Appendix B.2, jointly applying REPA and LSEP (rightmost) indeed results in the strongest class-wise and token-wise separability, which improves both training efficiency and generative performance. As noted in the Shared Update, this hybrid model further reduces FID by 1.2 points, achieving an FID of 4.7 (without CFG), and also improves FID under CFG from 1.42 to 1.37 compared to REPA alone, leading to SOTA performance, which is a very encouraging result for our approach.

---

> ### Author Response · Authors · 2025-11-21
>
> **(W3) Selecting the weight $\omega_{\textrm{class}}$.**
>
> Thank you for pointing this out. We observed that the primary denoising loss, $\mathcal L_{\textrm{velocity}}$, typically ranges from approximately 0.7 to 0.8 after a few iterations. In contrast, the regularization cross-entropy loss, $\mathcal L_{\textrm{class}}$, starts around 5 to 7 and gradually decreases, slowing down around 1.5 to 3. Since $\mathcal L_{\textrm{class}}$ is used as a regularization term, we chose its weight so that it contributes roughly 10\% of the denoising loss, corresponding to a target range of 0.07 to 0.08. Accordingly, we selected $\omega_{\textrm{class}}$ within the range [0.02, 0.05] to satisfy this condition.
>
> To validate this choice, we conducted an ablation study analyzing FID variations across different values of $\omega_{\textrm{class}}$ on the large model. We used $\rho_{L}=0.9$ and set the target depth to 7, and we did not apply random cropping to isolate the effect of $\omega_{\textrm{class}}$. This study confirmed that 0.03 is the most suitable value. The ablation results are provided in Appendix B.7 of the revised version.
> \
> \
> *Table. FID results for different values of $\omega_{\textrm{class}}$.*
> |**$\omega_{\textrm{class}}$**|**0.02**|**0.03**|**0.05**|
> |:--:|:--:|:--:|:--:|
> |**FID**|13.3|12.7|15.1|
>
> \
> **(W4) Scalability of LSEP to higher resolutions.**
>
> Thank you for your valuable feedback. We validated the scalability of LSEP on higher resolution ImageNet 512$\times$512, achieving an FID of 1.78, which outperforms REPA, which is a very encouraging result for our approach. We provide a detailed response in the “Shared Update’’ section and in Appendix D.
> \
> \
> **(Q1) Necessity of the time-dependent weighting of $\omega_{\textrm{class}}$.**
>
> We appreciate the opportunity to clarify this point. While the linear probe branch is conditioned on the timestep embedding, the classifier itself is not. Therefore, to classify intermediate features extracted under varying noise levels, we designed a time-dependent weighting schedule that updates the classifier based on time. This time-dependent weighting also assigns higher importance to higher-noise timesteps, helping the classifier maintain better linear separability even under large noise.
> \
> \
> **(Q2) LSEP performance trend under extended training (beyond 4M steps).**
>
> Beyond 4M steps, the FID exhibits minimal change. As noted in the "Shared Update," LSEP achieves the same FID as REPA at the corresponding epoch, and further training yields negligible improvement. Therefore, we report results at 4M steps to emphasize the efficiency of the training process.
> \
> \
> **(Q3) Generalization of LSEP to DiT models.**
>
> Thank you for this suggestion. Yes, LSEP can be applied to DiT models as well. Due to limited time and GPU resources, we could not conduct extensive experiments. However, initial experiments on DiT-L/2 with the following hyperparameters (Target depth: 8, $\rho_L = 0.9$, Random Crop: $n \in [12, 16] \cap \mathbb{Z}$, $\omega_{\textrm{class}} = [0.0025, 0.003]_{10\text{ steps}}$) showed improvement from a baseline FID of 23.3 to FID 20.3 at 400k iteraions (80 epochs). We have not included the DiT results in the revised submission yet, but we plan to add additional experiments and improved results in the final version.

---

> > ### Comment · Reviewer_bpbB · 2025-11-27
> >
> > I'd like to thank to the authors for the extensive rebuttal, the additional analyses, and the new experimental results within the limited rebuttal period. While the new experiments and justifications are helpful, some of my key concerns remain unresolved, and therefore I will keep my original score.
> >
> > - Linear separability as an objective
> >
> > I remain unconvinced that linear separability serves as a strong and general indicator of representation quality. The additional experiment using text prompts still appears limited in scope: converting ImageNet labels into templated prompts ultimately preserves a class-like structure rather than fully testing general text-conditioning. The performance under text conditioning remains weaker than class conditioning reinforces the concern that the approach may be suboptimal beyond class-based settings.
> >
> > - Sensitivity to $\omega_{\text{class}}$
> >
> > The observed sensitivity to $\omega_{\text{class}}$ further shows that the approach is fragile and dependent on careful tuning, rather than offering a stable or generally optimal strategy for representation learning.

---

> > > ### Author Response · Authors · 2025-12-02
> > >
> > > We appreciate the reviewer’s valuable comments.
> > >
> > > **(1) Questioning linear separability as a representation objective.**
> > >
> > > First, linear probing is a widely adopted and well-established tool for evaluating representation quality by measuring linear separability. Linear probing accuracy has been extensively validated in prior work as a meaningful indicator of how well semantic information is encoded in the learned features.
> > >
> > > We also agree with the reviewer that the additional text-conditioning experiment provided during the rebuttal does not fully test general-purpose text conditioning. Our goal in this experiment was not to claim comprehensive text-conditioning capability, but rather to extract text embeddings derived from class labels and use them to demonstrate the relationship between linear separability and training efficiency with different signal rather than class. In fact, as we already explained in our previous response, the concept of "linear separability" is not directly applicable to text prompts, since text prompts cannot naturally be separated into sets of points that are amenable to separation, unlike class labels. As we noted earlier these experiments were carried out to show, we are not directly dependent on the classification head and one-hot encoded classification labels, but our method of using linear representations extends to cosine similarity with text embeddings. However, for this experiment to make sense, we do still need an underlying set of points, whose linear separation makes sense, and this is why we generated the text prompts based on the class labels, as explained in our previous response.
> > >
> > >
> > > Furthermore, as also noted in our previous response, this experiment still uses hyperparameters tuned for the classification head, despite performing a different task of similarity with text embeddings. Therefore, its performance could be further improved with dedicated hyperparameter tuning. Nevertheless, even under these suboptimal hyperparameter settings, the text-conditioning setup already functions as an effective training regularizer and consistently improves generative performance, achieving an FID of 15.8, which outperforms the similar work, Diff.\& Disp. (arXiv:2506.09027) SiT-XL FID of 15.95 at 80 epochs. This supports our main claim that enhancing linear separability is beneficial beyond purely class-based supervision with a classification head.
> > >
> > > **(2) Regularization weight $\omega$ schedule**
> > >
> > > Many prior works that improve models through regularization require careful adjustment of the regularization coefficient for each model. Therefore, the sensitivity to $\omega_{class}$ is a natural characteristic of regularization-based methods and does not indicate a fundamental limitation of our approach. In practice, regularization coefficients are commonly tuned heuristically and vary across different architectures and model scales, with optimal values identified through grid search. Hence, the observed sensitivity to $\omega_{class}$ reflects standard tuning behavior rather than fragility specific to our method.
> > >
> > > We note that in similar works such as the SRA and Diffuse \& Disperse papers, which aim to improve training efficiency through self-alignment (SRA, arXiv:2505.02831) or self-contrastive learning (Diff. \& Disp.), the regularization coefficients are **similarly tuned heuristically**, varying across different models and model sizes, with optimal values identified via grid search.

---

### Official Review · Reviewer_xeMs · 2025-10-31

**Soundness:** 2
**Presentation:** 3
**Contribution:** 3
**Rating:** 4
**Confidence:** 4

**Summary:**

This paper proposes a regularization method, Linear SEParability (LSEP), to improve the training efficiency and generation quality of diffusion models, without relying on external pre-trained encoders.

The idea is to incorporate a linear classifier, inspired by linear probing in post-hoc evaluation, directly into diffusion training as an auxiliary objective, encouraging the model to learn more linearly separable intermediate representations. The approach is straightforward but also includes several specific training strategies and hyper-parameter choices.

Experiments on the ImageNet 256x256 dataset show that LSEP accelerates convergence and improves final performance across various model sizes of SiT using the original SD-VAE. It also demonstrates a synergistic effect when combined with existing alignment-based methods like REPA.

**Strengths:**

- The paper's primary strength lies in its novel and straightforward idea: a linear classifier can directly be an effective training regularizer. This provides a new, simple alternative to the dominant paradigm of external representation alignment.
- Although the idea of supervised training is simple, several tricks like patch-level random cropping are clever and effective.
- The empirical results are convincing. The method demonstrates consistent improvements over baselines. The finding that LSEP and REPA are complementary is also valuable.

**Weaknesses:**

- Training cost: The paper claims improvements in training efficiency, demonstrated by training iterations vs. FID scores.
  - However, the proposed method appears to require two separate forward passes through the first few layers, as the two branches require different class-conditioning (Figure 1, Section 3.2).
  - This could introduce significant computational overhead, potentially doubling the cost for a portion of the model, and this overhead is not discussed. A wall-clock time (or compute) vs. FID plot would be helpful.
- On linear separability: The paper is motivated by a claim that enhancing linear separability is the mechanism driving the improvements in generation. However, evidence for this link is sparse.
  - The main results in Table 1 do not report the linear probing accuracy for each configuration, making it impossible to directly correlate the degree of separability with the final FID. If we plot the (acc, FID) points for all 21 configurations in Table 1, will it demonstrate a clear, strong correlation?
  - In Table 2, it is also unclear how REPA and LSEP independently (and jointly) improve linear separability.
  - While Figure 4(a) shows an increase in accuracy, the absolute value remains relatively low (below 60%, much weaker than supervised learning), which warrants further discussion.
- Generalizability to other settings: The method, in its current form, explicitly relies on class labels for the auxiliary loss. This dependency limits its applicability to other tasks such as text-to-image or unconditional generation.
  - It would strengthen the paper to demonstrate how LSEP could be adapted for such scenarios, for instance, by training on class or attribute labels as a regularizer while still performing text-conditional or unconditional generation for the primary task.
  - Can the input conditions and labels for auxiliary classification be decoupled?
- Loss weighting design: Eq. (7) appears somewhat hand-crafted and might be sensitive to changes in the training-time timestep sampling (e.g., LogNorm distribution in SD3 and VA-VAE).
  - The ablation study in Table 1 shows that this parameter is carefully tailored to a very narrow range of values ([0.0275, 0.0325]).
  - The paper also does not show how the general magnitude of this value (around 0.03) is determined, as the cross entropy loss itself is already low enough (<0.1, according to Figure 6).

**Questions:**

Please refer to the weaknesses regarding the correlation between linear separability and performance, and the generalizability to other conditioning signals.

The current manuscript also reads like a strong empirical study with good performance but leaves many questions unanswered. For example, will other well-known techniques in supervised learning (e.g., label smoothing) be helpful? Will other discriminative tasks (other than a simple classification) also be beneficial? Why does a classifier trained on diffusion backbone lag behind supervised ViTs?

---

> ### Author Response · Authors · 2025-11-21
>
> We sincerely appreciate your thoughtful feedback and reviewer’s careful reading of our paper. We are pleased that the simplicity of using a linear classifier as a training regularizer and the effectiveness of our design choices, such as patch-level random cropping, were well received. We are also encouraged that the empirical consistency of our results and the complementary behavior of LSEP and REPA were recognized. We appreciate the opportunity to address the reviewer’s concerns.
> \
> \
> We would like to share several important updates based on additional analyses relevant to all reviewers. Please refer to the “Shared Update’’ section before reading our individual responses.
> \
> \
> **(W1) Computational overhead of dual-branch training and its impact on efficiency.**
>
> We appreciate the reviewer’s valuable comment and insightful suggestion. We agree that performing two separate forward passes up to the intermediate layers increases computational costs. However, the target intermediate layers in LSEP are relatively shallow, with 4 out of 12 in the Base model, 7 out of 24 in the Large model, and 8 out of 28 in the X-Large model, which does not significantly increase computation. Furthermore, LSEP does not require training a pretrained encoder, simplifying the process. In contrast, REPA requires training a large-scale model and also needs feature extraction from a pretrained encoder during diffusion model training, which reduces processing speed.
>
> Using 8 A100 GPUs with a global batch size of 256, the number of iterations processed per second is reported in the table below. Although there is a 4 to 13.4 percent decrease in speed compared to the baseline, LSEP achieves a FID of 7.54 at 160 epochs, outperforming the baseline's 8.3 FID at 1400 epochs, which demonstrates that the overall training efficiency remains high. Additionally, since the linear probe branch is not used during inference, it does not affect the inference time. This has been incorporated into Appendix B.6 of the revised manuscript.
> \
> \
> *Table. Number of iterations processed per second for different models using different methods.*
> |**Model**|**SiT(Baseline)**|**$\quad$REPA**|**$\quad$LSEP**|
> |:-------:|:---------------:|:------:|:------:|
> |B/2|8.07iter./s|7.11iter./s|7.75iter./s|
> |L/2|4.15iter./s|4.01iter./s|3.72iter./s|
> |XL/2|3.08iter./s|2.98iter./s|2.67iter./s|
>
> \
> **(W2, Q1) Further clarification on the link between linear separability and generation quality.**
>
> We thank the reviewer for raising these important points.
>
> **Linear probing accuracy in Tab. 1.**
> While we would have been happy to update linear probing results for all configurations, limited time and GPU resources made this infeasible during the rebuttal period. Instead, we implemented linear probing tests for each best-performing configuration (highlighted in dark color) within each component in Tab. 1. Linear probing was performed on SiT-L/2 models using each configuration at $t=0.1$. For feature extraction depth, we used 8 for the baseline and the selected target depth for the other variants. All experiments followed the same setup: a linear probe was trained on the ImageNet 1.28M training set for 90 epochs with a batch size of 8192, and evaluated on the 50K validation set.
>
> As shown in the table below, all methods improved by LSEP achieve higher linear probing accuracy than the baseline. Although the second row yields the highest linear probing accuracy, we note that accuracy generally increases as the target depth becomes deeper, since more layers contribute to linear separability. Under the same conditions (i.e., using the same target depth), accuracy consistently improves as each designed component is incrementally applied. This supports the observation that linear probing accuracy positively correlates with improvements in generation quality. This has been incorporated into Appendix B.5 of the revised manuscript.
> \
> \
> *Table. Linear probing evaluation on different configurations of SiT-L/2 models*
> |**$\rho_L$**|**Target**|**$\quad$ Rand.Crop.**|**$\qquad \quad \omega_{\textrm{class}}$**|**FID↓**|**sFID↓**|**IS↑**|**Pre↑**|**Rec↑**|**Acc.↑**|
> |:-----:|:--------:|:------------:|:---------:|:------:|:--------:|:-----:|:------:|:------:|:-------:|
> |  | |BaselineSiT-L/2 | |18.5|5.20|72.2|0.64|0.63|18.63|
> |0.9|8|n=16|0.03|12.9|5.37|96.2|0.67|0.64|58.05|
> |0.9|7|n=16|0.03|12.7|5.34|98.1|0.67|0.64|56.62|
> |0.9|7|n∈[12,16]∩ℤ|0.03|12.5|5.34|98.8|0.67|0.64|56.87|
> |0.9|7|n∈[12,16]∩ℤ|$[0.0275,0.0325]_{10\textrm{Bins}}$|12.3|5.40|98.3|0.68|0.64|57.43|

---

> ### Author Response · Authors · 2025-11-21
>
> **(W2, Q1) Further clarification on the link between linear separability and generation quality.**
>
> **Improving linear separability with combining REPA and LSEP in Tab. 2.**
> REPA and LSEP individually improve linear separability in different ways. REPA enhances representations by performing token-wise alignment using an external visual encoder, which primarily promotes patch-level representation quality. In contrast, LSEP improves linear separability by operating mainly on mean representation vectors.
>
> This suggests that combining the two methods should yield a more powerful representation geometry. As shown in Appendix B.2, jointly applying REPA and LSEP (rightmost) indeed results in the strongest class-wise and token-wise separability, which improves both training efficiency and generative performance. As noted in the "Shared Update", this hybrid model further reduces FID by 1.2 points, achieving an FID of 4.7 (without CFG), and also improves FID under CFG from 1.42 to 1.37 compared to REPA alone, leading to SOTA performance, which is a very encouraging result for our approach.
> \
> \
> **Relatively lower accuracy compared to supervised learning in Fig. 4(a).**
> The objectives and training strategies of standard supervised learning (i.e., classification) and the classification component in LSEP are fundamentally different. LSEP is not designed to optimize the classification task itself rather the classification objective functions as a lightweight regularizer that enhances linear separability while jointly training the denoising objective. To promote linear separability rather than classification performance, LSEP purposely uses a *linear* classifier without nonlinearities, whereas traditional supervised learning networks employ deep nonlinear classifiers. Although adding nonlinear heads could increase accuracy, it would contradict our design goal of using the probe to strictly asses linear separability of intermediate features as a regularizer rather than as a high-capacity classifier.
>
> Several structural differences further explain why LSEP exhibits lower classification accuracy than fully supervised models such as ViT, while remaining more suitable for generative training. (1) LSEP applies the linear probe only to a small subset of intermediate layers (up to 8 layers in SiT-XL/2), whereas models like ViT utilize all 24 (or more) transformer layers. (2) These intermediate layers in LSEP are simultaneously used for the denoising task, which constrains their capacity to specialize for classification. (3) LSEP does not employ data augmentation, which further contributes to the lower classification accuracy compared to supervised pipelines. We clarified these points in Appendix. B.8 of the revised manuscript.

---

> ### Author Response · Authors · 2025-11-21
>
> **(W3, Q1) Generalization to different conditionings (e.g., Text) or unconditional generation.**
>
> We thank the reviewer for this great question about a possible extension. This paper focused on class-conditional generation, where the concept of "linear separability" naturally makes sense among different classes. We use these conditions to enhance the linear separability of intermediate representations. In the text-to-image (T2I) case, the conditioning shifts from class labels to text features extracted from a text encoder such as CLIP. In this scenario, "linear separability" is hard to define since we cannot naturally group text prompts into sets that can be separated. Nonetheless, this raises an important question, and a potential expansion: Can we use a linear combination of the intermediate features to perform different tasks? For a T2I model, an appropriate evaluation target would be similarity with text embeddings.
>
> To this end, we conducted additional experiments during the rebuttal period to assess linear separability in text-based conditioning *without relying on class labels*. We converted ImageNet class labels into textual prompts of the form “A photo of [class]” and generated text-conditioned features using CLIP. The classifier head was removed, and a linear mapping-based text similarity head was introduced. This setup allows us to assess linear separability of intermediate features without actually training a classifier/linear probe, since objects remain distinguishable by class. Note in this case, our auxiliary task for assessing linear separability has shifted from a classification task on categorical variables to a similarity calculation task with continuous variables. LSEP can readily be applied in this setup with the appropriate changes.In practice, the text-conditioning model follows the same conditioning exposure strategy used in class-conditional training (arxiv:2209.12152, arXiv:2410.06940) utilizing conditional text features with high probability and unconditional text features with low probability for the purpose of CFG.
> \
> \
> *Table. Results on text conditioning training with LSEP*
> |**Method**|**Conditioning**|**FID**|
> |:--------:|:--------------:|:-----:|
> |**SiT-L/2(Baseline)**|Class|18.8|
> |**SiT-L/2+LSEP**|Class|12.3|
> |**SiT-L/2+LSEP**|Text|15.8|
>
>
> As shown in the table above, LSEP with text conditioning as defined in our on SiT-L/2 also improves the FID compared to the baseline. We note that we used the hyperparameters tuned for the classification head rather than those optimized for the new linear mapping–based text similarity. Consequently, the results under text conditioning are not directly comparable to those obtained with class conditioning. Incorporating text prompts to enhance linear separability further improves model training, demonstrating that LSEP is not inherently tied to a classification task and can be extended to broader tasks that operate on a linear combination of the network's intermediate features. The detailed experimental setup and 3D visualizations are provided in Appendix C of the revised manuscript.
>
>
> While we were unable to run full T2I experiments on large-scale models such as MMDiT, which would require a new study design, we consider this a promising direction for future research. We thank the reviewer again for suggesting this possible generalization of our idea.
>
> For the second question, yes, the input conditions and the labels used for auxiliary classification can be decoupled. For example, in the MS-COCO dataset, each image is paired with multi-class labels as well as a text prompt describing the image. The T2I model is trained conditioned on CLIP text features, while the class labels can be used separately for the cross entropy loss function in an auxiliary branch, following the LSEP framework, to enhance linear separability. Such extensions will be explored in the future work of LSEP.

---

> ### Author Response · Authors · 2025-11-21
>
> **(W4) Selecting the weight $\omega_{\textrm{class}}$.**
>
> We thank the reviewer for raising this question. We agree that it requires heuristic tuning and may need to be adjusted for different models that use different loss functions. Also, the value shown in Fig 6 corresponds to $\omega_{\textrm{class}} \times \mathcal L_{\mathrm{\textnormal{class}}}$ rather than the cross-entropy $\mathcal L_{\textrm{class}}$ itself. We apologize for the confusion and have corrected this in the revised version.
>
> When designing $\omega_{\textrm{class}}$, we observed that the primary denoising loss, $\mathcal L_{\textrm{velocity}}$, typically ranges from approximately 0.7 to 0.8 after a few iterations. In contrast, the regularization loss, $\mathcal L_{\textrm{class}}$, starts around 5 to 7 and gradually decreases, slowing down around 1.5 to 3. Since $\mathcal L_{\textrm{class}}$ serves as a regularization term, we selected its weight so that it contributes roughly 10\% of the denoising loss, corresponding to a target range of 0.07 to 0.08. Based on this, we chose $\omega_{\textrm{class}}$ within the range $[0.02, 0.05]$.
>
> To validate this choice, we conducted an ablation study analyzing FID variations across different values of $\omega_{\textrm{class}}$ on the large model. We used $\rho_{L}=0.9$ and set the target depth to 7, and we did not apply random cropping to isolate the effect of $\omega_{\textrm{class}}$. This study confirmed that 0.03 is the most suitable value. Building on this optimal value, we then empirically determined the time-dependent weighting schedule in the range $[0.0275,\ 0.0325]$. The ablation results are provided in Appendix B.7 of the revised version.
> \
> \
> *Table. FID results for different values of $\omega_{\textrm{class}}$.*
> |**$\omega_{\textrm{class}}$**|**0.02**|**0.03**|**0.05**|
> |:--:|:--:|:--:|:--:|
> |**FID**|13.3|12.7|15.1|
>
> \
> **(Q2) Potential extensions and limitations of the linear probe approach.**
>
> We thank the reviewer for raising these questions. In this study, our focus was not on improving the classifier itself, but on enhancing the linear separability of the intermediate features of the diffusion model using a simple single-layer linear classifier as a training regularization. Therefore, promising techniques such as label smoothing could be interesting and potentially helpful for the classification task, but we did not consider them as we followed the original linear probe method. For other discriminative tasks, LSEP can be extended by using different types of conditionings, such as segmentation or object detection, for the linear classifier, as discussed in (W3, Q1).
>
> Regarding why a classifier trained on the diffusion backbone lags behind supervised ViTs, we note that a *linear probe/classifier used here* is merely a single linear operator and does not include the multiple nonlinear activation layers that supervised models rely on. In addition, diffusion models differ fundamentally from supervised models in several aspects due to the need to coexist with the denoising objective. For further details, please refer to our response in (W2, Q1).

---

> > ### Comment · Reviewer_xeMs · 2025-11-25
> >
> > I would like to thank the authors for their detailed response and the significant effort put into the additional clarifications, particularly the computational overhead measurement, the linear separability correlation and the exploration of text-based regularization. However, several fundamental concerns remain unaddressed, which prevents me from raising my score:
> >
> > 1. **The approach is not elegant; several dedicated tricks are needed to mitigate the conflict as a workaround.** \
> > The rebuttal confirms that the generative and discriminative objectives are somewhat conflicted. The solution relies on a complex combination of engineering tricks: splitting branches with different conditioning probabilities (Sec 3.2), hand-crafted loss weighting (Sec 3.4), and the newly introduced exponential decay schedule from epoch 100. These adjustments appear to mask the underlying conflict rather than resolving it *fundamentally*.
> > 2. **The causality between linear separability and generation quality is not convincing.** \
> > The correlation between Acc and FID presented in Tables 5 & 6 is relatively weak. While we generally agree that better representations correlate with (or even lead to) better generation, linear probing accuracy is just one possible metric to reveal representation quality. We cannot say that explicitly optimizing Acc does *intrinsically cause* better generation quality. In addition, recent studies (arXiv:2505.10999) show that improved frameworks like Lightning-DiT can achieve higher LP Acc (62%) compared to LSEP’s optimized Acc (58%), despite different VAE spaces. This suggests that LSEP might bring more marginal benefits when applied to frameworks like this, since the approach is explicitly built on Acc, not the representation quality itself.
> > 3. **Generalization to non-classification head is a good direction, but remains preliminary.** \
> > I agree that better leveraging existing signals (rather than external distillation) is a right direction, but the current implementation via a classifier seems limited. In the new text-conditioning experiment (Table 9), the improvement drops when switching from classification (FID 12.3) to text-based similarity (FID 15.8). Moreover, this experiment might be better presented on a real text-conditional model. This indicates the generalization ability and the possible input-regularization decoupling warrant deeper research.

---

> > > ### Author Response · Authors · 2025-11-26
> > >
> > > We thank the reviewer again for the valuable feedback and for engaging in this discussion.
> > >  \
> > > \
> > > **1.**
> > > The elements highlighted by the reviewer are part of our methodological novelty rather than ad-hoc workarounds, and have clear reasonings behind them, as we will detail below. We first note that improving both the discriminative and generative aspects of diffusion models simultaneously has been fundamentally challenging because the two objectives inherently differ. Prior attempts to make these objectives coexist within a single diffusion model were ultimately unsuccessful, as the discriminative component degraded the generative performance (arXiv:2208.07791).
> > >
> > > In contrast, our approach was inspired by the observation, supported by alignment-based models such as REPA, that improving the representation quality in early layers benefits generation. While REPA relies on a pre-trained external encoder to achieve this effect, LSEP directly enhances the intermediate representations of the diffusion model itself and successfully addresses both (1) the coexistence of discriminative and generative behavior and (2) representation improvement using only the internal model through newly proposed mechanisms such as conditioning strategies and loss weighting.
> > >
> > >
> > > Finally, we respectfully note that the three novelties mentioned by the reviewer as engineering tricks are each driven by clear and carefully considered motivations: (1) To provide class conditioning for the main denoising task, we retained the conditioning used in the literature for the original denoising branch. As explained in the text, using the same conditioning information for the classifier branch creates a shortcut, where the classifier learns to classify based on the conditioning information rather than based on the intermediate features of the network. Thus, a different conditioning probability is *required* for this branch to make the generative and discriminative tasks work together. (2) Our linear classifier is a 2-layer linear network (by design). As a simple 2-layer network, there is no inherent time-embedding information in this network, yet it is evaluated across different noise levels in the generative model. Therefore, it is imperative that we weight classification on noisier images differently than classification on clean images. (3) A loss decay is helpful, because as we train, the intermediate features of the network become better, and performing linear classification gets easier. Thus, as we keep training, we want to put more weight on the denoising task and reduce the weight on the classification component, which is achieved by the proposed decay schedule.

---

> > > ### Author Response · Authors · 2025-11-26
> > >
> > > **2.**
> > > Linear probing accuracy can differ substantially depending on the noise level, the measured layer depth, and the design of the linear probing experiments (e.g., optimizer, batch size, normalization, etc.). In our work, we targeted early layers to allow the coexistence of classification and denoising tasks. If we had increased the depth from 7 to 10 in a SiT-L/2 as a target depth, linear probing accuracy would increase, but generative performance would decrease, due to the limited capacity of the model. However, as shown in Table 6, after identifying the optimal depth, our proposed methods improve linear probing accuracy while simultaneously enhancing generative performance. We are unsure whether the reviewer expects a relationship of LP Acc $\uparrow$ $\to$ FID $\downarrow$, regardless of the target depth, model, or other factors. This should not be the case, since as with all regularization schemes (used in training or in the broader optimization community), there are hyperparameters associated with the regularization term that will affect the performance of the regularizer. With our choice of hyperparameters, we clearly show regularizing the training with linear separability enhances generative performance.
> > >
> > > For the second part of this comment, the 62\% linear probing accuracy (FID 15.3 with CFG 1.5) reported for DiT-B in DDAE++ has several differences compared to our setup. First, they use a VA-VAE, which is already tuned via representation alignment to achieve stronger representations. Thus the different VAE spaces mentioned is arguably an advantage for DiT-B. Second, they extract different layers for linear probing accuracy, using 1) a noise level of 0.25, 2) the 10th layer out of 12, and 3) CFG = 1.5, whereas we use a noise level of 0.1, the 7th layer out of 24, and no CFG. Furthermore, the optimization techniques for linear probing training differ, including batch size, learning rate, and data augmentation.
> > >
> > > Finally, as the reviewer says, linear probing accuracy (or rather the linear separability of intermediate features) is one possible metric to reveal representation quality, which is the focus of this work. We are not claiming, it is the *only* metric that matters, but rather saying that its use as a training regularizer helps generation quality, which is verified with extensive experiments, both in the original submission and in the rebuttal that addresses the weaknesses initially raised by the reviewer.
> > > \
> > > \
> > > **3.**
> > > As we mentioned in the previous response, in the new text-conditioning experiment with LSEP, we used the hyperparameters tuned for the classification head rather than those optimized for the new linear mapping–based text similarity due to time constraints. Consequently, the results under text conditioning are not directly comparable to those obtained with class conditioning.
> > >
> > > We also did additional experiments to address the reviewer's comment about input-regularization decoupling. We kept the class conditioning of the SiT-L/2 model and applied the linear separability regularization term via text embeddings from CLIP. Specifically, we replaced the linear classifier with a linear mapping and changed the cross entropy loss to a cosine similarity loss. As a result, as shown in the table below, the FID improved from 18.8 to 14.6. Again, we highlight that this experiment still uses the hyperparameters tuned for the classification head albeit performing a different task. Yet even with sub-optimal hyperparameter settings, it serves as a training regularizer that improves generative performance. Finally, we agree that training a real T2I model with LSEP could further enhance our method, and we plan to explore this direction in future work. But we highlight that while these extension ideas build on the current work, their scope is well beyond this paper, and are more suitable for follow-up studies.
> > >
> > > |$\qquad$Method|Conditioning|$\quad$Regularization|FID|
> > > |:---:|:---:|:---:|:---:|
> > > |**SiT-L/2 (Baseline)**|Class|Class (C.E. loss)|18.8|
> > > |**SiT-L/2 + LSEP**|Class|Class (C.E. loss)|12.3|
> > > |**SiT-L/2 + LSEP**|*Class*|*Text* (Cos. Sim. loss)|*14.6*|
> > > |**SiT-L/2 + LSEP**|Text|Text (Cos. Sim. loss)|15.8|
> > >
> > > \
> > > \
> > > We hope this helps address your concern, and we would be happy to answer any additional questions you may have.

---

> > > > ### Comment · Reviewer_xeMs · 2025-11-26
> > > >
> > > > I thank the authors for their timely response and the additional, detailed clarifications. However, I still have reservations regarding the methodology and the underlying philosophy.
> > > >
> > > > **1. The weighting schedule could possibly have better solutions.** \
> > > > I acknowledge the clarification on the two-branch design to prevent a shortcut. However, the loss weighting (Sec 3.4 & exponential decay in rebuttal) remains a concern regarding its robustness and elegance.
> > > > - The reliance on specific heuristics (e.g., the 0.03 choice in {0.02, 0.03, 0.05}, or the [0.0275, 0.0325] bins) appears fragile, and this optimal schedule seems to be determined by searching with FID scores. If we switch to a different noise schedule such as LogNorm in SD3 or EDM, this schedule would likely need re-tuning.
> > > > - With that being said, some multi-task learning based loss weighting strategies might be helpful (e.g., the uncertainty-based multi-task learning method in EDM2), and could be more adaptive than the hand-crafted one.
> > > > - Additionally, the argument does not justify avoiding a time-dependent classifier. Designing a 2-layer probe with time-awareness (e.g., injecting a timestep embedding) would possibly be a more elegant solution to handle varying noise levels than manually engineering. It remains unclear why such a design was not adopted, given its potential benefits.
> > > >
> > > > *SD3: Scaling Rectified Flow Transformers for High-Resolution Image Synthesis* \
> > > > *EDM2: Analyzing and Improving the Training Dynamics of Diffusion Models*
> > > >
> > > > **2. Explicit classification vs. general representation learning:** \
> > > > The authors referenced prior work (arXiv:2208.07791) to highlight the difficulty of making classification and generation coexist. However, recent methods such as SRA, DDAE++ and "Diffuse and Disperse" successfully improve representation quality (as well as LP Acc, in SRA and DDAE++) using self-supervised approaches rather than explicit classification. This suggests that while enhancing representation quality is key, using a hard "classification" objective might be a suboptimal proxy (given the inherent difficulty noted by the authors themselves) compared to more general and "soft" self-supervised alignment.
> > > >
> > > > *SRA: No Other Representation Component Is Needed: Diffusion Transformers Can Provide Representation Guidance by Themselves* \
> > > > *DDAE++: Enhancing Diffusion Models Towards Unified Generative and Discriminative Learning* \
> > > > *Diffuse and Disperse: Image Generation with Representation Regularization*
> > > >
> > > > **3. Upper bound of the method:** \
> > > > My reference to the Lightning-DiT example previously was not intended to directly compare their LP Acc values against yours. I apologize if I did not make that clear. My question is regarding the "upper bound": If a future backbone inherently learns strong features that have a more saturated linear separability (e.g., >60% accuracy without LSEP, rather than the vanilla SiT's 18%), would applying LSEP still yield benefits?
> > > >
> > > > **4. The concept validation within the "linear separability" scope is still limited.** \
> > > > Setting aside the text-related discussion and sticking strictly to the scope of "linear separability", the proposed method seems only validated on a plain 1000-class cross-entropy classification, which does not fully cover the "linear separability" concept.
> > > > - Consider a dataset (e.g., a face dataset) with multiple co-existing attributes such as gender, age, glasses, or even person ID. It is unclear how LSEP would be formulated here. Would it require multiple linear heads?
> > > > - Consider a dataset (e.g., CIFAR-100) with multiple super-classes or clusters (even maybe pseudo-labels). Would LSEP be effective in this scenario, where input labels are different to the regularization classes?
> > > > - Furthermore, it is worth discussing whether other classification techniques, such as label smoothing or Supervised Contrastive Learning, would lead to more benefits than the current "hard" cross-entropy approach.
> > > >
> > > > Please do not misinterpret my comments. I do not mean to imply that the presented experiments are flawed. Rather, the current status of the paper leaves many follow-up questions regarding design choices, suggesting that further investigation is needed to present a fully comprehensive study within the current scope.

---

> > > > > ### Author Response · Authors · 2025-12-02
> > > > >
> > > > > **(2) Comparison with similar works (SRA, Diffuse and Disperse) and difference from DDAE++.**
> > > > >
> > > > > As mentioned in the Introduction of the main manuscript, prior works, such as SRA and Diffuse \& Disperse (Diff.\&Disp.), attempt to improve training efficiency using self-alignment (SRA) or self-contrastive learning (Diff.\&Disp.). Specifically, SRA aligns earlier layers with later layers, while Diff. \& Disp. extracts features from earlier layers and learns representations via contrastive learning. However, these methods remain fundamentally constrained by the representational capacity of the diffusion model itself, as they rely on feedback from their current representation abilities to guide further training. On the other hand, our method uses the linear separability of intermediate features as a means of quantifying representation quality, relying on a geometric characterization based on readily available conditioning signals, thereby improving both training efficiency and generative performance.
> > > > >
> > > > > As shown in the table below, SRA and Diff.\&Disp. lag significantly behind LSEP in terms of improvement at 80, 800, or even 1200 epochs. According to the reported scores, LSEP demonstrates notable gains in training efficiency across models under the same conditions. Importantly, while these prior models fail to reach the eventual performance of REPA, LSEP successfully achieves it.
> > > > >
> > > > > |Epoch|$\quad$$\qquad$SRA|$\qquad$Diff.\&Disp.|$\qquad$$\quad$LSEP|
> > > > > |:---:|:---:|:---:|:---:|
> > > > > |80|$\approx12.0$|15.95|**10.4**|
> > > > > |800|Notreported|7.71|**5.9**|
> > > > > |1200|-|7.43|-|
> > > > > |w/CFG|1.58(800epoch)|1.97(1200epoch)|**1.44**(800epoch)|
> > > > >
> > > > > \
> > > > > Furthermore, the aforementioned methods do not guarantee synergy when combined with REPA, because representations from a visual encoder would already geometrically cover the alignment. In contrast, LSEP employs a different methodology, allowing it to provide additional synergistic benefits, which we empirically validated across multiple experiments. This marks another major advantage of our method, as a training regularizer beyond representation alignment.
> > > > >
> > > > > Finally, DDAE++ has a different objective compared to the works mentioned above. Its goal is to enhance the representations of the diffusion model through self-conditioning while maintaining or slightly improving generative performance, making it suitable for use as a downstream encoder. For example, in diffusion transformers, DDAE++ improves representations in the later layers, but it does not primarily target generative performance or training efficiency. As shown in their paper, the generative results are only marginally improved compared to the baseline at the same number of training epochs.
> > > > > \
> > > > > \
> > > > > **(3) Upper bound of the LSEP method and clarification of LSEP benefits.**
> > > > >
> > > > > Yes, LSEP can still provide benefits. As mentioned in our previous response, the linear probing accuracy reported in the DDAE++ paper primarily reflects the *later* layers of diffusion transformers. As shown in Fig.4(a) of our manuscript, as well as in SRA and REPA, the later layers tend to saturate in terms of linear probing accuracy. Thus, methods like REPA or our regularization approach based on LSEP focus on improving representations in *earlier* layers (chosen as a hyperparameter), to further improve accuracy.
> > > > >
> > > > > Similarly, applying LSEP to a DDAE++ model would enhance linear separability in the *earlier layers*, improving overall linear probing accuracy across layers and boosting training efficiency. Therefore, LSEP remains a viable approach to further and more efficiently improve representation quality, even for backbones that already exhibit strong feature learning in the later layers.

---

> > > > > ### Author Response · Authors · 2025-12-02
> > > > >
> > > > > **(4) Extending LSEP to additional datasets and its advantages over REPA and experiments with label smoothing.**
> > > > >
> > > > > For datasets with multiple co-existing attributes, such as face datasets, we can adopt a multi-label classification approach. Specifically, we represent each attribute using a separate one-hot encoding vector and compute the binary cross-entropy loss for each attribute independently using a single classifier, as designed in our method. This encourages the intermediate representations to be linearly separable with respect to each attribute, allowing LSEP to enhance linear separability across multiple co-existing factors simultaneously. We anticipate that the higher level of differentiation using multiple attributes would further help LSEP, as we will detail next.
> > > > >
> > > > > Thank you for suggesting potential experiments on CIFAR-100. Following the reviewer's suggestion, we implemented the LSEP method on the CIFAR-100 dataset using SiT-L/2. We trained SiT-L/2 models for 4000 epochs under four settings: (i) SiT-L/2 with fine-grained 100 classes, (ii) SiT-L/2 with 20 super-classes, (iii) SiT-L/2 + LSEP with 100 classes, and (iv) SiT-L/2 + LSEP with 20 super-classes. This allowed us to investigate how the choice of conditioning clusters affects the training effects.
> > > > >
> > > > > As shown in the table below, fine-grained 100-class conditioning outperforms training with 20 superclasses for both the baseline SiT-L/2 and SiT-L/2 + LSEP. Notably, for both class settings, adding LSEP results in approximately a twofold speedup in training. These results demonstrate that training efficiency improves even when using superclasses, and that more detailed linear separability further enhances training performance, as one would expect. Overall, this provides clear evidence that improving linear separability has a direct and positive impact on the model’s learning efficiency.
> > > > >
> > > > > Another important point that these CIFAR-100 experiments highlight is the critical dependence of REPA on the input size, since it aligns features extracted from an external encoder. For example, to obtain representations from the visual encoder DINOv2, REPA resizes the input to $224 \times 224$, as the model is trained with $224 \times 224$ inputs and a patch size of $14$, which maps the input into $16 \times 16$ tokens. As a result, it can be aligned with a SiT model for $256 \times 256$ ImageNet using a patch size of $2$ in the latent space. However, for other resolutions such as $512 \times 512$ ImageNet, the input to the visual encoder must be resized to $448 \times 448$, and the positional embeddings must also be interpolated. In other words, additional preprocessing is required to extract representations. Moreover, for low-resolution inputs such as $32 \times 32$ (e.g., CIFAR-100) or $64 \times 64$ (e.g., $64$-resolution ImageNet), the visual encoder must be additionally trained, and thus REPA cannot be directly applied. In contrast, LSEP is completely independent of the input size and imposes no such constraints.
> > > > >
> > > > >
> > > > > |Method|$\quad$Num.ofClass|Epoch|FID|
> > > > > |:---:|:---:|:---:|:---:|
> > > > > |SiT-L/2|20(Superclass)|4000|24.42|
> > > > > |SiT-L/2+LSEP|20(Superclass)|2000|16.66|
> > > > > |SiT-L/2+LSEP|20(Superclass)|4000|**8.08**|
> > > > > |SiT-L/2|100(Fine-grained)|4000|4.65|
> > > > > |SiT-L/2+LSEP|100(Fine-grained)|2000|4.79|
> > > > > |SiT-L/2+LSEP|100(Fine-grained)|4000|**2.60**|
> > > > >
> > > > > \
> > > > > **Effect of label smoothing.** Following the reviewer’s suggestion, we conducted experiments using label smoothing for soft cross-entropy. We tested smoothing values of 0.1 and 0.05 on SiT-L/2 for 400K iterations. The resulting FIDs were 13.3 and 13.5, respectively, which did not outperform the hard cross-entropy baseline FID of 12.3, as one would expect, since label smoothing blurs out the class boundaries, making the additional gains from encouraging linear separability of intermediate features less prominent. While self-supervised or contrastive learning could be a potential direction to further improve our method, this is considered out of scope for the current paper and would constitute a separate project, as in prior works such as SRA (self-supervised) and Diff.\&Disp (self-contrastive).

---

> ### Author Response · Authors · 2025-12-02
>
> We sincerely thank the reviewer for the helpful and constructive feedback.
>
> **(1) Regularization weight $\omega$ schedule and classifier simplicity.**
>
> We agree that the coefficient of the regularization term ($\omega$) requires heuristic tuning. We believe that adjusting the coefficient to suit each model (e.g. a different noise schedule such as LogNorm in SD3 or EDM) is a natural requirement for regularization and does not indicate a limitation of our method. Indeed, in the SRA and Diffuse \& Disperse papers referenced by the reviewer, the regularization coefficients are *similarly tuned heuristically*, varying across different models and model sizes, with optimal values identified via grid search.
>
> Regarding the suggestion of a time-aware classifier, we thank the reviewer for this feedback. We also considered a time-dependent classifier and implemented a similar time-adaptive classifier for the linear probe following the protocol of U-REPA (arXiv:2503.18414). However, as shown in the table below, it did not yield improved results. While a time-aware classifier may help improve classification performance, we found it less effective than our proposed method for enhancing linear separability in the shared encoder. Therefore, we decided not to use a time-embedding classifier, which keeps our method consistent with the original linear probing approach while remaining simple and effective.
>
> |**$\qquad$$\qquad$$\qquad$Method**|**$\qquad$$\qquad$$\omega_{class}$**|**Iterations**|**FID**|
> |:---:|:---:|:---:|:---:|
> |**SiT-L/2+LSEP**|0.03|400K|12.7|
> |**SiT-L/2+LSEP (Time-aware Classifier)**|0.03|400K|13.3|
> |**SiT-L/2+LSEP**|$[0.0275,0.0325]_{10\textrm{bins}}$|400K|12.3|

---

### Official Review · Reviewer_BysX · 2025-11-01

**Soundness:** 2
**Presentation:** 3
**Contribution:** 1
**Rating:** 4
**Confidence:** 4

**Summary:**

This paper aims to accelerate the training of diffusion transformers without additional computational costs by using a linear classifier. Specifically, instead of aligning the intermediate features of the diffusion transformers with a pretrained vision encoder, it uses intermediate features to be linearly separable to classify the category of the given image. The authors demonstrate that the proposed method (LSEP) shows competitive results with REPA.

**Strengths:**

1. The idea is simple yet effective: It can even be used with REPA to further accelerate the training of the diffusion transformer.

2. Extensive analysis of the design choices facilitates the practitioners to use this approach.

**Weaknesses:**

1. LEAP is limited only to the dataset that has human annotations (i.e., labels). How can we use this method to accelerate training diffusion models that do not have ground-truth labels, e.g., T2I generation (MS-COCO)?

2. Even though REPA uses an external encoder as conditioning, the reported REPA results do not use any external labels (i.e., unsupervised), but LSEP needs labels (i.e., supervised). In some sense, LSEP uses more information to generate images, but shows worse performance and slower training than REPA (in terms of training time).

**Questions:**

1. Please answer the Weaknesses.

2. Table 2 shows SiT + REPA + LSEP. However, in Table 3, the authors report only SiT + LSEP. Can SiT + REPA + LSEP achieve SOTA performance?

---

> ### Author Response · Authors · 2025-11-21
>
> We sincerely appreciate your encouraging feedback. We are glad that the simplicity and effectiveness of our approach, including its ability to complement REPA and accelerate diffusion transformer training, were appreciated. We also value the recognition of our extensive analysis of design choices, which will be useful for practitioners.
> \
> \
> We would like to share several important updates based on additional analyses relevant to all reviewers. Please refer to the “Shared Update’’ section before reading our individual responses.
> \
> \
> **(W1, Q1) Generalization to Different Conditionings (e.g., Text).**
>
> We thank the reviewer for this great question about a possible extension. This paper focused on class-conditional generation, where the concept of "linear separability" naturally makes sense among different classes. We use these conditions to enhance the linear separability of intermediate representations. In the text-to-image (T2I) case, the conditioning shifts from class labels to text features extracted from a text encoder such as CLIP.  In this scenario, "linear separability" is hard to define since we cannot naturally group text prompts into sets that can be separated. Nonetheless, this raises an important question, and a potential expansion: Can we use a linear combination of the intermediate features to perform different tasks? For a T2I model, an appropriate evaluation target would be similarity
> with text embeddings.
>
> To this end, we conducted additional experiments during the rebuttal period to assess linear separability in text-based conditioning *without relying on class labels*. We converted ImageNet class labels into textual prompts of the form “A photo of [class]” and generated text-conditioned features using CLIP. The classifier head was removed, and a linear mapping-based text similarity head was introduced. This setup allows us to assess linear separability of intermediate features without actually training a classifier/linear probe, since objects remain distinguishable by class. Note in this case, our auxiliary task for assessing linear separability has shifted from a classification task on categorical variables to a similarity calculation task with continuous variables. LSEP can readily be applied in this setup with the appropriate changes. In practice, the text-conditioning model follows the same conditioning exposure strategy used in class-conditional training (arxiv:2209.12152, arXiv:2410.06940) utilizing conditional text features with high probability and unconditional text features with low probability for the purpose of CFG.
> \
> \
> *Table. Results on text conditioning training with LSEP.*
> |**Method**|**Conditioning**|**FID**|
> |:--------:|:--------------:|:-----:|
> |**SiT-L/2(Baseline)**|Class|18.8|
> |**SiT-L/2+LSEP**|Class|12.3|
> |**SiT-L/2+LSEP**|Text|15.8|
>
> As shown in the table above, LSEP with text conditioning as defined in our on SiT-L/2 also improves the FID compared to the baseline. We note that we used the hyperparameters tuned for the classification head rather than those optimized for the new linear mapping–based text similarity. Consequently, the results under text conditioning are not directly comparable to those obtained with class conditioning. Incorporating text prompts to enhance linear separability further improves model training, demonstrating that LSEP is not inherently tied to a classification task and can be extended to broader tasks that operate on a linear combination of the network's intermediate features. The detailed experimental setup and 3D visualizations are provided in Appendix C of the revised manuscript.
>
> While we were unable to run full T2I experiments on large-scale models such as MMDiT, which would require a new study design, we consider this a promising direction for future research. We thank the reviewer again for suggesting this possible generalization of our idea.

---

> ### Author Response · Authors · 2025-11-21
>
> **(W2, Q1) LSEP trains slower than REPA despite using external labels/more information.**
>
> Thank you for raising this point. We would like to clarify the differences between REPA and our method. Alignment-based models such as REPA rely on an external, large-scale pretrained encoder. For example, DINOv2 (Large) consists of 307M parameters with 24 transformer layers and is trained on the massive LVD-142M dataset using a self-supervised objective that yields stronger representations than supervised training. REPA leverages these powerful pretrained features by aligning the diffusion model’s intermediate representations with the encoder’s features, which provides strong representation guidance right from the beginning of training. However, training or obtaining such large-scale encoders requires substantial computational resources and large-scale datasets. Thus, REPA inherently benefits from a significantly richer amount of information distilled through large-scale pretrained encoders.
>
> In contrast, LSEP does not rely on external pretrained encoders and therefore does not receive strong representation guidance at the beginning of training. This naturally leads to slower improvement compared to REPA (while still being much faster than the baseline) during early epochs. Moreover, LSEP is not designed to optimize the classification task, rather the classification objective serves as a regularizer that enhances linear separability while jointly optimizing the denoising objective. Nevertheless, after sufficient training (approximately 400 epochs), LSEP matches the FID of REPA and continues to improve, as shown in the updated Fig 1.
> \
> \
> **(Q2) Results with CFG on SiT-XL/2 combining REPA and LSEP in Tab 3.**
>
> We thank the reviewer for raising this important point. As mentioned in the "Shared Update", combining REPA and LSEP improves the FID under CFG from 1.42 (REPA alone) to 1.37, leading to SOTA performance, which is a very encouraging result for our approach.

---

### Official Review · Reviewer_iiBB · 2025-11-02

**Soundness:** 3
**Presentation:** 3
**Contribution:** 3
**Rating:** 6
**Confidence:** 4

**Summary:**

- The paper proposes LSEP, a simple regularization that inserts a trainable linear probe at an intermediate layer of a diffusion transformer and co-optimizes it with the standard denoising loss, with the goal of directly promoting linear separability of intermediate features during training rather than measuring it post-hoc.
- On ImageNet 256×256 with SiT backbones, LSEP reduces FID versus the SiT baseline, improves linear-probe accuracy at early layers, and shows complementarity with alignment methods like REPA
- The paper positions LSEP as a no-alignment alternative that can also be combined with alignment (e.g., REPA) to further speed training and improve generation quality, and it discusses limitations such as the lack of experiments on text-to-image, video, or higher resolutions.

**Strengths:**

- The paper tackles a concrete and important problem of improving the quality of features learnt by generative models.
- The intervention is architecturally minimal and easy to adopt, since it adds a single linear head and combines it with the standard denoising loss via a simple weight.
- The experiments show consistent FID gains over baseline SiT and demonstrate complementarity with REPA, with clear tables and figures that place the numbers in context and avoid relying solely on a single configuration.
- The writing is generally clear about the two-branch setup and provides a useful hyperparameter grid for reproduction.

**Weaknesses:**

- The goal of pretraining is to learn general-purpose, task-agnostic features that transfer across datasets, label spaces, and objectives (e.g., detection, segmentation, retrieval, open-set recognition), but this method explicitly supervises a linear probe on ImageNet class labels during pretraining and pressures intermediate features to be separable with respect to that specific taxonomy. As a result, the learned geometry is optimized for class-conditional ImageNet generation and linearly decoding those same labels, which does not guarantee that the features capture fine-grained structure needed by unrelated downstream tasks and may bias representations toward dataset-specific shortcuts.
- The probe and denoiser use different class-conditioning schedules, which creates gradient-level tension; the paper notes that $ρ_L=1$ causes a mismatch but does not analyze interference between the two schedules or test mitigations such as stop-gradient into shared embeddings or separate class embeddings.
- The paper trains its linear probe on pooled features (and sometimes pooled crops), which can make the average representation linearly separable while leaving individual patch tokens messy. Without token-level evaluations, the claim that LSEP improves patch-space geometry remains unclear and could reflect pooling shortcuts

- The paper sets the probe’s unconditioning probability very high but below one, yet it does not test whether the method remains stable under label noise or whether those occasional non-null conditioning steps make the probe exploit conditioning artifacts rather than visual content. It will be interesting to see if the reported gains survive mislabeled classes and still hold when the class-conditioning pathway is perturbed or ablated.

**Questions:**

See weaknesses above

---

> ### Author Response · Authors · 2025-11-21
>
> We sincerely thank the reviewer for the thoughtful and encouraging feedback. We are glad that the importance of improving feature quality in generative models and the simplicity of our approach using a single linear head were positively received. We also appreciate the recognition of our comprehensive experiments, including consistent FID improvements and the demonstrated complementarity with the align-based method (REPA). We are grateful for the opportunity to clarify our contributions and address the raised concerns.
> \
> \
> We would like to share several important updates based on additional analyses relevant to all reviewers. Please refer to the “Shared Update’’ section before reading our individual responses.
> \
> \
> **(W1) Potential dataset-specific bias from class-supervised pretraining.**
>
> Thank you for this valuable comment. We agree with the reviewer that pretraining has broader purposes. However, our focus here is not on pretraining, but rather on training diffusion models on a given dataset, without targeting additional downstream tasks. Therefore, in this context, we believe that leveraging the additional taxonomy (e.g., class labels in the ImageNet setting) about the training database in fact aligns with our goal. Since class-conditional diffusion models readily operate with explicit conditioning, encouraging intermediate features to be linearly separable with respect to the given conditioning is consistent with our focus.
>
> We would also like to note that diffusion models trained on a particular dataset without LSEP are already influenced by the intrinsic dataset-specific biases of the training data. Nevertheless, as shown in Fig 4(b), the baseline model exhibits only weak linear separability even on the training dataset. In contrast, LSEP intentionally strengthens the linear separability guided by the provided conditioning signals, allowing the model to learn more structured and informative intermediate representations. While these intermediate representations are evaluated on a particular dataset, they may nonetheless enhance the model's ability to transfer across datasets and downstream tasks, thereby making its representations more versatile. Though we couldn't directly evaluate this during the rebuttal, it would be an interesting future research direction aligned with the reviewer's overall concern.
> \
> \
> **(W2) Gradient interference from differing class-conditioning schedules.**
>
> We thank the reviewer for raising this important point. When $\rho_{L} = 1$, the linear probe branch receives only unconditional labels (i.e., 100\% null class), which makes it focus almost entirely on promoting linear separability in the intermediate layers. Under this extreme setting, the denoising and linear probe objectives indeed exhibit gradient interference because they operate on shared intermediate features while using mismatched conditioning.
>
> Instead of analyzing gradients numerically, we studied this interference through a systematic ablation on the conditioning exposure probability $\rho_{L}$. Our results in Tab. 1 show that exposing the linear probe branch to a moderate fraction of class-conditioning strikes the right balance: it avoids the mismatch caused by $\rho_L = 1$ and enables both tasks to coexist effectively. These ablations provide an effective empirical analysis of when gradient interference becomes problematic and how it can be mitigated.
>
> We did not consider incorporating a stop-gradient mechanism, as the intermediate features must contribute to both denoising and separability, and the conditioning signals should also influence both objectives. Nevertheless, designing a stop-gradient variant could be an interesting extension.

---

> ### Author Response · Authors · 2025-11-21
>
> **(W3) Token-level evaluation of LSEP patch linear separability.**
>
> We thank the reviewer for pointing this out and apologize for not making it clearer. We agree that training the linear probe on pooled features does not guarantee improved linear separability for individual patch tokens. To address this, we employ random cropping during training as the position and size of the crops are randomly varied to form subsets, and the average of these cropped features is used by the linear probe. In this setting, the cropped features constitute a subset of the original features, encompassing a randomly chosen portion between 56\% and 100\% when $n \in [12, 16] \cap \mathbb{Z}$.
>
> For evaluation of patch-level linear separability, as described in Sec. 3.3, we visualize *each patch feature* using 3D projections. These visualizations show that random cropping improves the organization of patch features within each sample, demonstrating enhanced separability at the token level.
> \
> \
> **(W4) The stability of LSEP under perturbed or noisy class-conditioning.**
>
> We appreciate the reviewer’s insightful suggestion regarding testing the stability of LSEP under perturbed or noisy class-conditioning. We agree that evaluating whether improvements truly stem from enhanced visual representations rather than reliance on conditioning artifacts is important.
>
> To assess robustness under perturbed conditioning, we conducted experiments by intentionally injecting label noise. Specifically, we randomly mismatched 5\% and 10\% of the 1.28M ImageNet labels and trained SiT-L/2 models using the same hyperparameters as in Tab. 4.
>
> As shown in the below table, the FID increases as the proportion of mismatched labels grows, which is expected since noisy supervision degrades the formation of clean linear separability. However, LSEP still consistently outperforms the baseline even under 10\% label noise. Importantly, the linear probing accuracy exhibits only a slight degradation. This suggests that while noisy labels introduce a weaker separability signal, LSEP mainly strengthens the intrinsic visual separability and thus remains robust under label noise. These experiments, including the 3D t-SNE projection results, are presented in Appendix B.4.
> \
> \
> *Table. Evaluation of LSEP under perturbed class-conditioning.*
> |**Method**|**Noisy Level**|**FID**|**Linear Probing Acc.**|
> |:--------:|:-------------:|:-----:|:--------------------:|
> |**SiT-L/2 (Baseline)**|0%|18.5|18.63|
> |**SiT-L/2+LSEP**|0%|12.3|57.43|
> |**SiT-L/2+LSEP**|5%|14.4|55.66|
> |**SiT-L/2+LSEP**|10%|16.3|56.23|

---

> > ### Comment · Reviewer_iiBB · 2025-11-26
> >
> > Thank you for the detailed rebuttal and the additional experiments. These strengthen the paper's narrative and address most of my concerns.

---

### Author Response · Authors · 2025-11-21

We sincerely thank all reviewers for their valuable and constructive feedback. The rebuttal process provided an excellent opportunity to further strengthen and refine our work. Before addressing individual comments, we would like to share several important updates based on additional analyses and experiments conducted after the initial submission.
\
\
**1. Updates on LSEP performance.**

We have conducted additional investigations and experiments on our proposed method. We found that incorporating the classification loss $\mathcal L_{\text{class}}$ with a decaying weight is highly beneficial, especially in long-horizon diffusion training. Specifically, we applied an exponential decay to the classification loss starting from epoch 100, *while keeping all other methods (configurations) exactly the same as reported in the initial paper.* Using this decayed weighting for the classification loss, our single-model setup achieves the same FID score (5.9) as REPA, which relies on a large-scale visual encoder, at 800 training epochs. In addition, with classifier-free guidance (CFG), our method attains an FID of 1.44, comparable to REPA's 1.42.
\
\
Furthermore, we trained a combined approach of REPA and LSEP up to 800 epochs. Remarkably, this hybrid model matches the 800 epochs REPA performance at just 400 epochs. At 800 epochs, it further reduces FID by 1.2 points, achieving an FID of 4.7 (without CFG), and improves the FID under CFG from 1.42 to 1.37. Both results establish new state-of-the-art performance compared to REPA, which is a very encouraging result for our approach. We provide the updated results in Fig. 1, Tab. 1, and Tab. 2.
\
\
**2. The scalability of LSEP on ImageNet 512$\times$512.**

We additionally verified the scalability of LSEP at the higher ImageNet 512$\times$512 resolution, where it achieves an FID of 1.78, outperforming REPA's FID of 2.08. Due to the limited time available during the rebuttal period, we followed the fine-tuning strategy used in DDT (arxiv:2504.05741), starting from a mid-training checkpoint at $256 \times 256$. Specifically, we used the 400-epoch checkpoint of the SiT-XL model trained at $256 \times 256$ and fine-tuned it for an additional 40 epochs using the same configuration as in Tab. 4, except that we adjusted the random cropping range to $n \in [24, 32] \cap \mathbb{Z}$ to match the increased latent feature size of $32 \times 32$ and a batch size of 512.

This demonstrates that LSEP scales effectively to higher resolutions. These results have been incorporated in Appendix D of the revised manuscript, and we plan to include results from training from scratch as well as fine-tuning with additional epochs in the final version.
\
\
**3. Summary of Changes.**
- Fig. 1, Tab. 1, and Tab. 2 are updated with new results in the main manuscript.
- Appendix B.4: Stability of LSEP under perturbed (or noisy) class-conditioning.
- Appendix B.5: Linear probing evaluation on different LSEP configurations.
- Appendix B.6: Analysis of computational speed.
- Appendix B.7: Ablation study on $\omega_{\textrm{\textnormal{class}}}$.
- Appendix B.8: Differences between the classifier in LSEP and supervised learning models.
- Appendix C: Text conditioning in diffusion transformer training with LSEP.
- Appendix D: Experiments on ImageNet at a resolution of 512~$\times$~512.

\
We hope these shared updates further clarify our contributions. We now respond to each reviewer individually.

Thank you once again for your valuable time and expertise.
\
\
Warm regards,

Authors of submission 20427

---

### Author Response · Authors · 2025-12-03
**Final Remarks**

We would like to sincerely thank the Reviewers, Area Chairs, Senior Area Chairs, and Program Chairs for the time, expertise, and constructive feedback devoted to our submission. We are encouraged by the reviewers’ positive recognition of our work, commending the novelty of simplicity and effectiveness of our approach to accelerate diffusion transformer training (**all reviewers**). The reviewers also highlighted its ability to complement REPA (**Reviewers iiBB, BysX, and xeMs**). In addition, the comprehensiveness and consistency of our experimental results and ablation studies were recognized as effectively validating our design choices (**all reviewers**). The usefulness of our analysis for practitioners were also appreciated (**Reviewer BysX**).
\
\
During the rebuttal, we provided substantial additional experiments and analyses to address the concerns raised by the reviewers, focusing mainly on:
\
\
**Superior performance of LSEP *without relying on external encoders*.**

Our proposed method, LSEP serves as a broadly applicable training regularizer and achieves the same FID score (5.9) as REPA at the same 800 training epochs on $256 \times 256$ ImageNet, despite not using any large-scale external visual encoder. With classifier-free guidance (CFG), LSEP attains an FID of 1.44, which is comparable to REPA’s 1.42. Furthermore, on $512 \times 512$ ImageNet, LSEP achieves an FID of 1.66, outperforming REPA’s FID of 2.08, demonstrating its scalability. These results achieve SOTA performance, which is a highly encouraging result for our approach. (Fig. 1, Tab. 1, and Appendix D)

In addition, LSEP can be applied to inputs of any resolution, whereas REPA is limited to specific input sizes (e.g., $256$ and $512$) due to its dependence on an external encoder pre-trained on those resolutions. For example, as we showed during the rebuttal, LSEP can be directly applied to CIFAR-100 with $32 \times 32$ inputs, significantly improving both training efficiency and generative performance. In contrast, applying REPA to CIFAR-100 would require additional training of a separate external visual encoder. (Appendix E)
\
\
**Linear separability as a crucial factor for accelerating and improving diffusion model training.**

We demonstrate in the manuscript, through various experiments such as linear probing evaluation and 3D feature projection, that the linear separability of intermediate features plays a key role in accelerating and improving diffusion model training. In addition, following the reviewers’ suggestions, we have included extensive new experiments to further strengthen the paper, including: an analysis of the stability of LSEP under perturbed (or noisy) class conditioning (Appendix B.4); linear probing tests on different proposed configurations (Appendix B.5); experiments with different conditioning signal (e.g., text) using LSEP (Appendix C); and a comparison of the effects between superclass and coarse class on CIFAR-100 (Appendix E). All these experiments consistently indicate that improving linear separability of intermediate features directly contributes to enhanced generative performance.
\
\
**Additional experiments**, such as an ablation study on the regularization coefficient ($\omega_{\text{class}}$) and an analysis of computational speed, were conducted following the reviewers' requests. Clarification on the differences between the classifier in LSEP and those in supervised learning models was also provided. Please refer to the "Shared Update" for details.
\
\
Consequently, Reviewer iiBB indicated that their concerns had been addressed. Unfortunately, we were unable to engage in further discussion with Reviewers BysX, xeMs, and bpbB due to the OpenReview incident. However, we comprehensively addressed their comments in our rebuttal and believe that the provided clarifications resolve the raised points.
\
\
We appreciate the constructive engagement from all involved in this process and hope the final decision will take into account the positive developments and strengthened evidence resulting from the discussion phase.
\
\
Warm regards,

Authors of submission 20427

---

### Meta-Review · Area_Chair_PCtT · 2026-01-07

**Summary:**

the main concerns were about “is this really a solid, general idea or just a tuned recipe?”

Reviewers liked the simplicity and strong ImageNet results, but several felt the method depends on multiple interacting tricks (branch split, weighting schedule, decay) that may need re-tuning elsewhere.

Some weren’t convinced linear separability is the real cause of better FID, versus just a correlation. Others questioned how well it extends beyond class-labeled setups (real text-to-image, unlabeled data), and whether comparisons to REPA are fully fair given different supervision and compute assumptions.

Based on the reviewer feedback and the author response, the area chair recommends rejection.

**Reviewer Concerns:**

Rebuttal clearly fixed the “missing evidence” type issues: added strong numbers (LSEP matching REPA at equal epochs, hybrid REPA+LSEP improving further), higher-res (512) results, speed/overhead measurements, weight ablations, label-noise robustness, CIFAR-100 superclass vs fine-grained tests, and even tried label smoothing and a time-aware classifier.

Still outstanding: xeMs and bpbB stayed unconvinced on the bigger picture: the method feels over-engineered and maybe fragile, the linear-separability-to-generation story still isn’t "causal" enough, and the "generalizes beyond class labels" evidence feels preliminary (text experiments still basically preserve class structure).

**Reviewer Scores:**

Reviewer iiBB likely bumps up since they said the rebuttal addressed most concerns.
BysX probably nudges up a bit because the rebuttal answered their main questions (hybrid SOTA, clarifying REPA vs LSEP tradeoffs, some text-based extension).
xeMs likely stays at 4: they repeatedly said the core philosophy and robustness issues block a raise.
bpbB also likely stays at 4: they explicitly kept their score and remained skeptical about linear separability as a general objective and the sensitivity to the weight.

---

### Decision · Program_Chairs · 2026-01-26

Reject